



# Insights from 20 years of temperature parallel measurements in Mauritius around the turn of the 20th Century

Samuel. O. Awe[1], Martin Mahony[2], Edley Michaud[3], Conor Murphy[1], Simon J. Noone[1], Victor K.C. Venema[4], Thomas G. Thorne[5], Peter W. Thorne[1*]

[1]Irish Climate Analysis and Research Units (ICARUS), Department of Geography, Maynooth University, Maynooth, Ireland
[2]School of Environmental Sciences, University of East Anglia, Norwich, NR4 7TJ, UK
[3] Mauritius Meteorological Society, Quatres Bornes, Mauritius
[4]Meteorological Institute, University of Bonn, Germany
[5]Independed researcher, Co. Kildare, Ireland

*Correspondence to*: Peter W. Thorne (peter@peter-thorne.net)



**Abstract.** There is considerable import in creating more complete, better understood, holdings of early meteorological data. Such data permit an improved understanding of climate variability and long-term changes. Early records are particularly incomplete in the tropics, with implications for estimates of global and regional temperature. There is also a relatively low level of scientific understanding of how these measurements were made and, as a result, of their homogeneity and comparability to more modern techniques and measurements. Herein we describe and analyse a newly rescued set of long-term, up to six-way parallel measurements, undertaken over 1884-1903 in Mauritius, an island situated in the southern Indian Ocean. Data include: i) measurements from a well-ventilated room, ii) a shaded Thermograph; iii) instruments housed in a manner broadly equivalent to a modern Stevenson Screen; iv) a set of measurements by a Hygrometer mounted in a Stevenson Screen; and for a very much shorter period v) two additional Stevenson Screen configurations. All measurements were undertaken within roughly 80 metre radius. To our knowledge this is the first such multidecadal multi-instrument assessment of meteorological instrument transition impacts ever undertaken, providing potentially unique insights. The intercomparison also considers the impact of different ways of deriving daily and monthly averages. The long-term comparison is sufficient to robustly characterise systematic offsets between all the instruments and seasonally varying impacts. Differences between all techniques range from tenths of a degree Celsius to in excess of a degree Celsius and are considerably larger for maximum and minimum temperatures than for means or averages. Systematic differences of several tenths of a degree also exist for the different ways of deriving average / mean temperatures. All differences bar two average temperature series pairs are significant at the 0.01 level using a paired t-test. Given that all thermometers were regularly calibrated against a primary Kew standard thermometer this analysis highlights significant impacts of instrument exposure, housing, siting and measurement practices in early meteorological records. These results reaffirm the importance of thoroughly assessing the homogeneity of early meteorological records.



## 1 Introduction

The adoption of the Paris Agreement, with its focus upon efforts to keep global surface temperature warming below 2 degrees Celsius (°C) above pre-industrial levels and strive to remain below 1.5 °C has led to a renewed focus upon early instrumental records given their importance in establishing estimates
of the pre-industrial baseline and thus how close we are to these levels of warming (e.g. Hawkins et al., 2017). These early records were undertaken by a range of individuals and groups using a broad variety of instrumentation, exposures, practices and even temperature scales (Knowles Middleton, 1966; Parker, 1994; Venema et al., 2020). In the early 20th Century the advent of the International Meteorological Organization (IMO, the precursor to the modern day World Meteorological
Organization (WMO)) strengthened the push to standardisation of surface meteorological station instrumentation and observing practices at national, regional and international levels and the expansion of networks to be more geographically complete. This served to greatly improve spatio-temporal comparability of these latter measurements. From a climate monitoring perspective, the 'raw' early records suffer from potentially complex biases that are heterogeneous in nature and occurred across a
geographically sparse network (Hartmann et al., 2013 and references therein).

When such biases change they can cause inhomogeneities in climate series, together with changes in the surroundings and relocations of stations, leading to errors in estimates of long-term climatic changes and trends. Thus, prior to use in climate applications these records must be assessed for homogeneity
and adjusted to remove apparent data artefacts. Modern state-of-the-art techniques generally use comparisons between nearby stations to identify and then adjust for non-climatic data artefacts (Conrad and Pollak, 1950; Aguilar et al., 2003; Trewin, 2010; Menne and Williams, 2009; Venema et al., 2020). The breakpoint detection step is inherently a signal-to-noise (SNR) issue - the lower the noise in the series the smaller the breakpoints which can be robustly detected - which means the comparator stations
should be close enough to yield a difference series with low variability (Williams et al., 2012; Gubler et al., 2017; Lindau and Venema, 2018a). Unfortunately, station sparsity is a particular issue for the tropics, continental interiors, and much of the southern hemisphere in the early record. Figure 1 shows available early tropical records (defined as 30°N to 30°S) secured to date by the Copernicus Climate Change Service and the National Oceanic and Atmospheric Administration's National Centers for
Environmental Information (NOAA NCEI) in putting together a new integrated set of land data holdings (Thorne et al., 2017, Noone et al. 2021). There are vast swathes of the tropics with no information in current archives and stations drop in and out, which may speak more to data rescue progress and data policy issues than to the lack of potential long-term records (Allan et al., 2011).

However, most studies on homogenization have been developed on and verified / validated against relatively dense networks representative for temperature observations made during recent decades in Europe and the USA, with some notable exceptions. Gubler et al. (2017) studied the influence of station density by comparing homogenization outcomes using all Swiss temperature stations to homogenization with a thinned-out network with a similar network density to that found in Peru. They found that when
the network was thinned out the homogenization method HOMER could improve the homogeneity at a station level, but could not reduce the network average trend errors, which is the most crucial task.



Chimani et al. (2018) studied the homogenization of relative humidity observations in Austria, which have much lower cross-correlations between stations than temperature and found that none of the homogenization techniques could achieve clear improvements in the homogeneity of the data. Lindau and Venema (2018a) helped understand the problem. They found that when the SNR is too low, the errors in the positions of break inhomogeneities are very large. Small errors in the break positions can lead to drastic under-adjustment of any network-wide trend biases due to inhomogeneities (Lindau and Venema, 2018b). The latter was also found by Williams et al. (2012) in their benchmarking exercise for the USA, which used scenarios of varying difficulty for the inhomogeneities. For easy scenarios well over 90% of the network-wide trend errors could be removed by homogenization, but for the hardest scenario with many small breaks only about half of the trend error could be removed.

For early records, which are much sparser, it is considerably less certain to what extent such approaches can be fully effective. A variety of other techniques have been used to assess these early records, many of which rely upon intra-station characteristics, available parallel measurements, or meteorological covariates to adjust for apparent inhomogeneities (Hubbard et al., 2001; Camuffo,2002; Böhm et al., 2010; Brunet et al. 2011; Auchmann and Brönnimann 2012; Lindén et al., 2015; Kaspar et al. 2016; Kaspar et al. 2016; Acquaotta et al., 2016).

Knowledge of early instrumental set-ups and observing practices at individual sites is highly varied. Tropical and subtropical sites are known to have used a broad variety of approaches that generally, but far from exclusively, were some form of thatched, pagoda screens, or sheds, or well-ventilated rooms (Parker, 1994). Outside of the tropics, window screens and garden stands were used. The basic scientific premise was to expose the thermometer to the ambient atmospheric conditions whilst simultaneously avoiding direct exposure to solar radiation. This was achieved with varying degrees of success. Figure 2 illustrates a number of tropical / sub-tropical exposures in Parker (1994) and uncovered in the present analysis via personal contacts. This shows the existence of significant heterogeneity in instrumental exposure in these early records. For most stations, these early designs were replaced by the Stevenson (1864) Screen (also known as Cotton Region Shelter), which is much more enclosed and aims to maintain good ventilation, but provide better protection against indirect sun and infra-red radiation. In general, changing to Stevenson Screens tended to reduce recorded temperatures; the diurnal and annual cycle of these biases suggest this was mostly by reducing radiation errors.

These historical transitions, changes in observational methods that happened all over the world, could give rise to substantial global trend biases, but are also especially difficult to homogenize because not only the candidate, but also the comparator stations are generally affected. How difficult this problem is depends on the percentage of stations that are affected quasi-simultaneously in a given region and period, how fast the change took place, and how much the size of the break varies, for example because it depends on the local climate or particularities of the set-up at each site. Recognizing their importance quasi-contemporaneous breaks were included in benchmarking exercises (Venema et al., 2012; Williams et al., 2012), but not studied in much detail. Gubler et al. (2017) found that the breakpoints due to automation of the Swiss network around 1980 were often not detected and thus not corrected if they used a sparse station network. In the MULTITEST benchmarking two scenarios have network-



wide breaks introduced over two years for half of the network or over 30 years for all of the network (Domonkos et al., 2021). They found that homogenization could improve the homogeneity of the data at a station level, but only the homogenization methods ACMANT and PHA were able to significantly reduce network-wide trend biases and only by about a quarter. We thus have reason to believe that a
low SNR is especially a problem for statistical homogenization when it comes to network-wide similar breaks, such as the introduction of the Stevenson Screen around the turn of the 20th Century.

Ultimately, however, trust in statistical and metadata based techniques can only come from a fundamental measurement-based understanding. There has thus been increasing interest in recent years
in ascertaining the differences arising from the systematic changes in instrumentation and methods of observation via direct comparison of modern and old instruments, exposures and practices. To date these have primarily considered the impact on early mid-latitude records. For a site at Kremsmünster, Austria, the original instrumentation has remained, permitting a side by side comparison in the modern era (Bohm et al., 2010). For Spanish data, early screens were reconstructed from available metadata and
a comparison undertaken focusing on the effects of exposure differences (Brunet et al., 2011). These studies have highlighted important and seasonally varying impacts in these specific cases. Ongoing are three known analyses: i) an experiment by the Netherlands Met Service KNMI comparing a Pagoda housed instrument to a Stevenson screen using a similar philosophy to that in Brunet et al.; ii) analysis of 60 years of parallel measurements in Adelaide, Australia (Blair Trewin, pers. comm.); and iii) two
comparisons of North wall measurements with Stevenson screens in Norway (Nordli, pers. comm.). There have also been several long-running sets of parallel measurements associated with far more recent transitions, such as the US transition from Cotton Region Shelters to the MMTS (Maximum-Minimum Temperature Sensor) sensors with a side by side comparison now exceeding 30 years (figure in Cook, 2016). But there exist very few, if any, known contemporaneous such experiments for the
earliest instrumental transitions. It does not, however, follow that no such efforts were undertaken. Parallel measurements tend to have not been shared as widely as long-term station records and efforts are ongoing to build a database of such measurements (Venema et al., 2018).

Herein we recover and analyse just such a recently rediscovered contemporaneous set of parallel
temperature measurements which was undertaken for a period of 20 years at the Royal Alfred Observatory in Mauritius around the turn of the 20th Century. Perhaps uniquely, this set of measurements consists of up to 6 independent sets of observations using distinct methods of observation. This enables a much more robust assessment than a typical long-term two-way comparison or short-term intense intercomparison campaign, permitting greater insights. The remainder of the
analysis is structured as follows: Section 2 provides a brief history of the observatory and introduces the context of the parallel measurements series. Section 3 describes the rescue and collation of the records by the lead author as part of his Masters thesis. Section 4 analyses the set of parallel measurements for annual and seasonal effects and the impacts of monthly averaging choices. Section 5 provides a discussion and Section 6 concludes.



## 2 History of meteorological observations on Mauritius, the Royal Alfred Observatory, and specifics of the experiment

### 2.1 History and key personnel involved

Mauritius is a remote island location in the southern Indian Ocean (20°10" South, 57° 31" East) (Fig.
3). Given its strategic position, colonial control of Mauritius was highly contested with control changing
hands from the Dutch (1598-1710), to the French (1710-1810) and then the British (1810-1968) before
gaining independence. Mauritius has played a key, but often unrecognised, role in the development of
meteorological research. A comprehensive review is given in Mahony (2018) with a focus on
understanding of tropical cyclones, and is based upon extensive archival research. The interested reader
should refer to that work for a fuller exposition of the historical role of meteorology in the history of
modern-day Mauritius than is possible here. A brief history of meteorological research in Mauritius is
also given by the National Meteorological Service at http://metservice.intnet.mu/about-us/historical-
background/.

The very first temperature measurements on Mauritius using thermometers were made by Mr Cere of
the Jardin du Roy (now Botanical Gardens) in Pamplemousses in 1774 under a shaded office veranda.
Later, Mr Lislet Geoffroy, an engineer, astronomer, botanist and cartographer, started measurements in
Port Louis with a thermometer under his resident veranda and as a scientist he began publishing his
observations locally from 1830. In 1832 a Public Observatory was opened in Port Louis on the wharf in
the Harbour and started temperature measurements on a 12 hourly basis, morning and afternoon. The
Royal Engineers Observatory in 1852 also made such observations on a regular basis (Henry James
Scheme) until 1856 when all instruments were transferred to the Public Observatory due to the
demolition of the Royal Engineers Observatory. The Observations continued in Port Louis until 1869
when a new site was located in Pamplemousses to build the Royal Alfred Observatory in 1870 and
operations commenced in November 1874 under Dr Meldrum's directorship.

The Royal Alfred Observatory (RAO henceforth) plays a pivotal role in the overall history of Mauritian
meteorology over the late nineteenth and early twentieth Centuries. The RAO was conceived in 1860
and opened in the early 1870s, attaining the status of a government department in 1874. The
observatory provided meteorological services to the then colony and oversaw the gradual development
of a network of observing sites across Mauritius from the beginning of the twentieth century. The
meteorological service headquarters was relocated in 1925, although observations continued at the site.
The building was eventually closed in 1961 and pulled down to make way for the construction of a
hospital. Figure 4 shows a contemporary photo of the building and surroundings. The two sets of annual
reports of the observatory imaged by NOAA NCEI (Section 3) highlight a broad range of measurements
being undertaken which may be of interest to many investigators. In addition to meteorological
measurements there are, for example, some measurements of ozone reported in at least some of the
'blue book' series (Section 3) and the meteorological observations are collated alongside magnetic
observations after 1898.



Brief pen sketches of key personnel who may have had a role in the measurement program being analysed herein are as follows:

**Charles Meldrum** arrived in Mauritius in 1848 to teach mathematics at the Royal College, having
previously spent a number of years teaching in Bombay. In 1851 he helped launch a Meteorological Society and in 1861 he was made Government Observer. He was Director of the RAO from its opening in 1874 until 1896, so it most likely would have been him who started, or at least approved, the experiment. His big interest, and that which he is now mainly known for, was 'cyclonology'. He spent much of his time as Society secretary compiling data from ships' logs and stitching together pictures of
cyclones in the Southern Indian Ocean (many of these records are preserved in the annual logs used herein). With these, he was able to confirm the hypothesis that the winds blow in a spiral towards the centre – 'Meldrum's Rules' for navigating in a storm thereafter became a key reference for mariners. Like many at the time he was committed to cyclical understandings of climatic variability, and keenly sought out correlations between sunspots and local weather. He also took an interest in the broader
environmental state of the island, particularly the relationships between climate, deforestation, disease and population growth. Again, this interest is clearly reflected in the composition of the annual summary logbooks.

While his major contributions to meteorology (e.g. those for which he was made Fellow of the Royal
Society in 1876) did not necessarily come from observatory-based work, we have evidence from the 1850s of a very 'modern' concern for careful and precise instrumentation. This was not to be taken for granted at the time – the officer Meldrum replaced as Government Observer was using his own homemade instruments, for instance. Further details on Charles Meldrum are given in Michaud (2000).

**Thomas F. Claxton** – slightly less is known about Claxton, beyond a few publications and some second-hand information from Albert Walter, his successor (see below). He had been a computer at Royal Greenwich Observatory from 1890, having previously won prizes at school in mathematics, geometrical drawing, navigation, astronomy and French. He was recruited from Greenwich by Meldrum in 1895, to the post of First Assistant Director and Director Designate. When Claxton took over as
Director in 1896, Albert Walter was likewise recruited from the Greenwich ranks. Claxton published a little on cyclones and astronomy, and was clearly more interested in the latter than Meldrum had been. He seems to have been very mathematically-minded and a precise observer, albeit not such an effective Director – he frequently created trouble with volunteer observers on outlying islands, as well as quarrelling frequently with the government and the press over cyclone warnings. While Meldrum was
known as a great 'savant' and a trusted forecaster, Claxton was regularly accused of missing approaching cyclones (this was 'single-station forecasting', an art developed in part on the basis of Meldrum's earlier theoretical work), and in 1910 he left Mauritius for a position in Hong Kong.

**Albert Walter** – in the meantime, Walter had developed a reputation as a somewhat more reliable
cyclone forecaster. He arrived in 1897 and spent much of his first year studying up on Meldrum's cyclone work. From around 1900 onwards he took up some of Meldrum's statistical interests, beginning work on the relationships between cyclones and the sugar crop, the island's chief export (Rouphail,





2019). He argued that by using statistical interpolation it was possible to infer with some accuracy, based on only a couple of point observations of wind, how much cane each estate would have lost with a passing cyclone, and more accurately than a visiting insurance inspector could. He was much more statistically-minded (climatological perhaps) than Claxton, and much more embedded in the life of the
colony. He married into the French landowning elite, served their interests much more directly, and ended up doing varied statistical work for the government alongside his eventual RAO Director duties. Indeed, when he eventually left for East Africa in 1925, it was to conduct statistical work primarily, although he did end up establishing a meteorological service there from 1929. Walter's later career was very much a product of the interwar burst of interest in agricultural meteorology.

Less can be said about the assistants who may well have been doing much of the day-to-day observing. We have found passing references to a few of them – mostly they were recruited from the Royal College, and trained in making observations by the Director or Assistant Director over a period of a few weeks. If they were deemed good enough, they'd be kept on. They seem to have been mostly drawn
from what was known as the 'creole' community, which at that point meant French descent, but born in Mauritius, although some of Indian descent were also recruited. How work was divided up, and who was perceived as being capable of what, was very racialised. One of the assistants who worked in the period of the experiment, a Mr Figon, retired early in 1899 on account of the ill effects (including failing eyesight) of the routine observational work, which included spending a lot of time in the
basement with photographic chemicals. His case, along with that of a number of assistants lost to malaria, is indicative of how physically demanding observatory work could be in this period, especially in tropical environments.

## 2.2 Particulars of the temperature intercomparison at the Royal Alfred Observatory

The parallel measurements documentation is a little ambiguous and scattered across the so-called blue-
book series (1884-1909 but with gaps in the versions available from NOAA NCEI archives) and annual reports (1887-1973) as detailed in Table 1. Early reports contain only superficial metadata. The first annual summary report from the observatory imaged at NOAA NCEI (Section 3) mentions annual temperature series taken since 1875 (the first full year of operation) and compares 1887 to the 1875 to 1887 average (Fig. 5). Given that the initial observatory was based upon a visit by Charles Meldrum to
the Royal Greenwich Observatory in 1866 to both acquaint himself with the layout and acquire initial instruments, it can be assumed that the instrumentation and set-up would have been broadly similar to that in use there. Details of instrumentation at Kew and advocated for use elsewhere under UK auspices are given in a Royal Society report (Royal Society, 1868). Interestingly, this report alludes to a period of parallel measurements carried out at Kew in 1867 which may have been the inspiration for the longer
running program of parallel measurements at RAO.

Correspondence dated 30th March 1875 states that an anemometer and a Thermograph (Kew pattern, by Adie) had been ordered, but that the Thermograph was yet to arrive. The Thermograph seems to have arrived in December 1875, but as stated by Meldrum in his annual report drafted in Oct 1876, "this
instrument has not been mounted, owing to the want of a building for it". The Blue Book report for



1876 repeats that the Thermograph is still not yet working, and refers to the Casella maximum-minimum thermometers having a southern exposure, and being mounted 6 feet from the ground. A building which was under construction for the Thermograph seems to have been damaged in a storm, further delaying the installation. The use of Casella thermometers with a southern exposure would be methodologically quasi-consistent with the earliest measurements on Mauritius taken under verandas (Section 2.1).

It is clear across the series of reports that there were at least four distinct instrument set-ups in operation, for periods of at least several years each, across varying subsets of the period 1883-1903:

1. In a well-ventilated room situated between open south and east facing windows which some reports refer to as 'the main (or principal) computing room'
2. From a Thermograph located in what is variously described as a 'shed' or in latter reports a Photoheliograph dome and thermograph room.
3. From a Stevenson Screen located on the lawn
4. From a Hygrometer located in a Stevenson Screen on the lawn

For details of the instrumentation and locations the 1899 'blue book' report (and then repeated in subsequent reports with some variations) contains quite detailed metadata about both the relative positioning and manufacturers of much of the instrumentation as follows (with minimal editing):

- The main building was built in 1875 and is described as facing north by west, and as a stone structure of rectangular base 56 feet long by 38 wide (17x11.6 m) and sheltered on all sides by a two storied veranda. On the ground floor are three rooms with the principal computing room on the eastern side. In the south east corner of this room are self-registering maximum and minimum thermometers, dry bulb (Casella Nos. 15447 and 1470) and wet bulb (Casella Nos. 958 and 1464); they are between two open windows, one facing the east and the other the south. The 1900 report mentions that these were mounted on a wooden stand. This appears to constitute the original long-standing series referred to in the 1887 report. Latter reports suggest replacement of the thermometers may have been necessary as the identifiers either change or are not stated. Also, earlier reports lack such detail so instrument replacements in the prior period cannot be ruled out.
- The Photoheliograph Dome and Thermograph Room are a stone building built in 1878 some 240 feet (73m) NE of the main building which is 16ft (4.9 m) in diameter. Adjoining are two rooms, one for photographic operations and the second of which, the 'east room', contains the registering parts of the Kew thermograph and was known as the Thermograph Room. The "stems" of the photographic thermometers projected southward into the Thermograph screen, being held in position by a metal frame, which was also attached to the standard dry and wet bulb thermometers. The screen was 6 feet square by 6 to 7 feet high (1.83x1.83x2.13 m), with the roof sloping towards the south. The sides were double louvre boarding, and the planks of the floor were double with an air space between to ensure ventilation as well as protection against radiation from the ground. The bulbs of the thermometers were two feet (0.6 m) above the floor and 6 feet (1.83m) above the ground (although the 1891 annual report implies they are 6 feet 4 inches above the ground). As far as can be ascertained from available metadata, the thermograph





was a photographic type by the well-known manufacturer James Hicks (although as noted below a further thermograph was ordered a decade prior). The 1902 'blue book' describes the thermograph as of the Kew pattern being fully described in the *Annual Report of the Meteorological Committee for the Royal Society for the Year 1867* (Royal Society, 1868) to which the interested reader is referred for full details of this instrumentation. The recorder was located in the main building. Lenses were used to reflect the temperature / mercury level in the tube and the image was shown on a specially prepared chart and later developed on a scale. The recorded temperature was therefore known on the next day.

- The enclosure (we can only assume this means fenced area although the metadata is not specific) for thermometers was at a distance of about 40 yards (36.5m) to the east of the main building in a circular enclosure 11 yards (10m) in diameter. In it were placed: 1. A Stevenson Screen containing self-registering maximum and minimum thermometers (Negretti and Zambra Nos. 40,450 and 40,467) which the 1891 annual report states is 4 feet (1.2m) above the ground; and 2. A Stevenson Screen containing a Mason Hygrometer by Casella (thermometers nos. 81524 and 81525) being 4 feet (1.2m) above the ground, amongst a range of additional meteorological equipment. Available metadata is insufficient to ascertain to what extent the Stevenson Screen set up may be similar to or deviate from a modern Stevenson Screen set-up.

There is no metadata to suggest that any of these instrumentation combinations or locations changed over the period of record being considered here, although several thermometers are recorded as being replaced. Figure 4 implies that there is little in the way of relief at the observatory and its immediate surrounds, which means that all measurements are probably undertaken at broadly equivalent elevations although precise elevation details are only given for the principal computing room, which hosts the barometer. The sun elevation is climatologically to the north of the Mauritius location except for a two month period from mid-November to mid-January, so the south facing position of the well-ventilated room and thermograph serve to minimise potential solar radiation impacts.

Additional buildings are noted in latter blue books as follows:
- A magnetic observatory constructed in 1874 situated 60 yards (55m) to the north of the building and measuring 40 ft long and 34ft wide (12.2x10.4m).
- A small wooden building built in 1875 for absolute magnetic observations situated 80 yards (73m) to the NW of the main building and 60 yards (55m) to the west of the magnetic observatory.
- About midway between the Photoheliograph Dome and the magnetic observatory is a Seismograph room, a wooden building of 12 ft square (3.7mx3.7m) and some 18ft (5.5m) high built in 1894.
- Some 16 yards (14.6m) to the south of the Seismograph Room is a small wooden hut erected in 1885 for the Balfour Stewart Actinometer.

A sketch of the layout based upon this information is provided in Fig. 6, which also denotes the assumed approximate location of the photo given in Fig. 4. It is apparent from this sketch that other built infrastructure than that in which the instrumentation were variously housed is unlikely to have had a substantial material effect on the series.



From the metadata contained in the 1900 blue book, which contains specific dates of commencement for the Stevenson Screen based measurements, in combination with the data being reported in each set of year books and blue-books, the following timeline can be deduced: i) that the measurements in the principal control room (well-ventilated room) in the main building commenced in 1875 and until the installation of the Thermograph (timing uncertain) were considered the principal measurement series; ii) that on Feb 1st 1883 the Stevenson Screen was installed; iii) tentatively that the Thermograph started being explicitly used in reports from 1891, although the 1884 blue book talks about temperatures being by photography implying the Thermograph may have been in use at that time (note that metadata in the early blue books extends to one page whereas later editions extend to ten or more pages) and the 1887 annual report alludes to new thermometers being sourced for the Thermograph. However, the 1888 annual report specifically states that the principal temperature records given relate to the well-ventilated room-based measurements. The Thermograph had a maximum and minimum thermometer added 16th June 1900; and iv) that the Hygrometer screen-based measurements started 16th August 1891, although there is some doubt on this timing as a set of annual Hygrometer screen-based measurements (screen type unstated) is given in the 1890 blue book report (see Section 3).

There are several allusions to other instrumentation. The 1887 annual report mentions a cage on the lawn being distinct from the Stevenson Screen, but in the annual average measuring almost identical temperatures. Extreme daily values are reported for each month for the cage and the well-ventilated room-based measurement in that report, but not the monthly averages. The only monthly averages reported arise from the Thermograph. The cage is again mentioned in the 1888 annual report, and then the 1894 report, but not thereafter. For an unspecified period of time starting in April 1892 an experiment with the Stevenson Screens was undertaken. Two additional Screens were erected - one at 6.5 feet high and the other of larger dimensions as laid out on p. 4 of the 1892 annual report (2 ft 4 in long, 15 in broad and 2 ft 10 in high – 71 cm x 38 cm x 86 cm), also set at 4ft (1.2 m) high and compared to the long-standing Stevenson Screen. The fact that the new screen was considered large perhaps points to the long-standing screen being considerably smaller than modern Stevenson Screens which may point to potential housing heating and ventilation issues in comparison to modern equivalent equipment. Data from this experiment were presented in reports only for April to December 1892. It is unclear whether the experiment persisted beyond 1892 or not and, if so, whether the data still remain.

Numerous reports point to the use of a suite of primary standard thermometers to perform regular calibration. A Kew standard 107, which had been constructed in 1853, was replaced in 1891 according to the annual report with a standard Kew thermometer No. 701 following cross-characterisation. These standard thermometers were used in the early period monthly and then latterly at least twice a year to calibrate the various instruments, including a number of additional thermometers, which were used to calibrate a range of other instrumentation, assuring comparability. These comparisons are tabulated in the Blue book entries from 1901 forwards, but are not reported prior to this. The calibration is quite involved and would have been state-of-the-art at the time (Fig. 7). The adjustments were applied to the reported series such that any differences in the annual reports relate principally to siting, radiation shielding or observational practices such as times of observation. The 1903 blue book report discusses





comparison with a new standard sent from Kew. Various reports mention this series of Kew standards also being used to characterise a range of additional thermometers on the island and on passing ships.

The 1901 blue book, starting on page xxc (30) and running over some 7 pages includes a significant
analysis related to shortcomings of the maximum thermometer installed in the Thermograph. An in-depth tabular comparison is shown that was too substantial to digitise in the current work. Results for the Thermograph maximum temperatures over June-1900 to August-1901 should be accordingly treated with a degree of caution. Some of the issue appears to relate to the initial choice to position the thermometers horizontally instead of vertically.

It is somewhat unclear what happened after the cessation of the parallel measurements program or why the program was ceased, although reports leading up to 1903 increasingly lament the poor wages, the challenges of reaching the observatory, working conditions, and the rates of sickness, which seemingly increased sharply. The 1904 and 1905 blue book meteorological reports state unequivocally that:
'*Observations of air and evaporation temperature in the Principal Computing Room, and in a Stevenson Screen on the lawn, were discontinued on 1903 December 31'* implying that the Thermograph was retained as it is this instrument that continues to be described in the metadata descriptive front matter. Whereas other sources suggest that the operational measurements switched to being made using a Stevenson Screen in a dedicated meteorological plot at the RAO in 1903. If so, it is
unclear whether this Stevenson Screen is the same screen as that used in the long running inter-comparison or a new screen was commissioned and whether the 'dedicated plot' was the same enclosure. Reports from 1904 onwards from a non-exhaustive review consistently mention only one set of temperatures, so even if parallel measurements continued in some form they are not reported in the available reports.

**3 Data discovery and digitisation of monthly data**

The lead author, Samuel Awe, expressed an interest in undertaking data rescue and analysis for his MSc in Climate Change at Maynooth University. The corresponding author, Peter Thorne, pointed him to the holdings that were imaged by NOAA NCEI in the summer of 2012 over a period of about three months. Considerable effort had been made by NCEI to index the international meteorological holdings lodged
with them and preserved in their physical archives (Fig. 8). Identified records of interest were then imaged to the extent resources permitted and these images were hosted as tar files at ftp://ftp.ncdc.noaa.gov/pub/data/globaldatabank/daily/stage0/FDL/. To date, little to no exploitation of these images has occurred, and it is unclear to what extent the imaged data represent data already digitised or which remain to be digitised. The holdings have yet to be integrated fully into the WMO
Data Rescue database to our knowledge at this time.

The original aim of the research was to identify a set of holdings, which may extend a record in a remote location of the globe to provide the largest possible increment to our understanding of historical climate change from a Masters thesis. Via cross-checking with the ISTI databank (Rennie et al., 2014),





and the then secured holdings of the Copernicus Climate Change Service Global Land and Marine Observations Database contract, it was identified that the early holdings at Mauritius imaged by NCEI were either entirely undigitized or yet to be integrated into these holdings (at any of monthly, daily or synoptic resolutions). Subsequent investigations have highlighted that discontinuous records are

included in the CRUTEMv5 product (Osborn et al., 2021). These images could augment existing digital holdings held by NOAA NCEI which extended back only to the middle of the 20[th] Century. There are two directories of images for this location that are summarised in Table 1. Both pertain to the Royal Alfred Observatory. The original intent was simply to mine these repositories and digitise relevant meteorological data to extend the data records as far back as possible. The presented reports vary

stylistically throughout the period of record. The annual summary reports contain a wealth of metadata, meteorological data and observations on agricultural production and disease. Additional meteorological parameters include pressure, humidity, wind and rainfall. More broadly these reports provide very valuable insights upon both contemporary science and society on the island at this time. The blue book series are more data rich and contain many observations to daily and report level granularity. Later blue

books incorporate additional magnetic observations and many contain annexes.

As the work progressed the interesting intercomparison of temperature measurement methods was discovered. This led to a reprioritisation of the work to concentrate upon this valuable long-term series of parallel measurements. The monthly parallel measurements digitised and analysed herein are

contained in annual summary sheets as shown for example in Fig. 9 as well as the blue books. The paper records available in imaged form from NOAA NCEI, supplemented by two absent blue books sourced from the NOAA Foreign Data Library, provide results from the parallel measurements over a combined period of 1884 to 1903 (Fig. 10). Data availability depends upon whether the annual reports and the blue books were available or just one or the other. The reports start with a period from 1884 to

1889 of parallel measurements solely between the Stevenson Screen and the thermometers in the well-ventilated room, but with 1886 missing as neither report type is available (the sole year of such an occurrence). Starting in 1890 the Hygrometer in a Stevenson Screen begins to be reported and from 1891 the Thermograph temperature measurements appear in the reports. Thermograph, Stevenson Screen and well-ventilated room-based measurements then continue through the end of 1903. The

Stevenson Screen experiment measurements make a brief appearance in 1893. The report format varies through time. What is tabulated varies by instrument and through time with the Thermograph generally containing the most data. Some instruments are only ever recorded as monthly summaries. Temperatures were digitised in the originally reported Fahrenheit scale and subsequently converted to Celsius as part of the analysis. All monthly-resolution intercomparison data were digitised by S. Awe

and P. Thorne and are available at https://doi.pangaea.de/10.1594/PANGAEA.935684 (original units) and https://doi.pangaea.de/10.1594/PANGAEA.935683 (converted to Celsius).



## 4 Analysis of the rescued parallel measurements

### 4.1 Maximum and minimum temperatures

Maximum and minimum temperatures (Tx and Tn respectively henceforth) are available for a substantive period of overlap across the three principal measurement series (Fig. 10). Were the blue

books available from other sources for 1886 it is very likely that almost 20 years of continuous records could be reconstructed for the well-ventilated room-based and Stevenson Screen records. Even with this gap there exist almost 19 years of records from the well-ventilated room and Stevenson Screen and 13 years from the Thermograph – more than sufficient to enable a robust comparison of these series.

The Tx timeseries are clearly distinct from one another (Fig. 11) with the Stevenson Screen reading warmest and the well-ventilated room coolest. Differences are substantial being of the order 3°C between the well-ventilated room and the Stevenson Screen and with the Thermograph about half way between once it becomes available. The differences are broadly stable throughout the period of record, with the exception of the well-ventilated room-Stevenson Screen series which appears to have a

relatively small step change associated with the break in data availability in early 1892 (just after the Thermograph series appears) and also a marked change in seasonality between 1884-85 and all subsequent years. To study the change in seasonality in the early record there are only the two instruments available, and combined with the cursory metadata recorded, it is impossible to ascertain why this may have occurred. For the shift around 1892, there is for most of 1891 and thereafter also the

Thermograph available permitting a 3-way comparison. The room-Thermograph series shifts more than the thermograph-Stevenson series and in the same direction as the well-ventilated room-Stevenson series suggesting that this shift may principally relate to a change in the well-ventilated room measurements. It is possible that the major cyclone of April 1892 may have impacted operations, although this event arises in the period when the reports, relative to later volumes, contained scant

metadata and there is no obvious record of the impacts from the material to hand. Overall, the short period and the temporary break in Stevenson Screen measurement series availability precludes a robust quantification and assignment of the break to one or other of the well-ventilated room or the Stevenson Screen.

Seasonality of the Tx differences (Fig. 12) exhibits a marked annual cycle for the well-ventilated room minus Thermograph comparison. The difference in Tx between the well-ventilated room and the Thermograph peaks in Austral summer and is a minimum in Austral winter. For the two remaining comparisons involving the Stevenson Screen there is relatively little obvious structure to the differences. For all months the differences in all comparisons are non-zero distributed. All paired Tx

differences are highly significant under a paired t-test (Table 2).

The Tn timeseries are, again, clearly distinct from one another (Fig. 13) but now with the Stevenson Screen reading coolest and the well-ventilated room the warmest. There are no obvious breaks in the apparent behaviour of the difference series between instruments. The opposite sign of the differences to

those for Tx means that differences between instruments in Tx and Tn are much larger than in Tm



(Section 4.2). This is consistent with the contention in Thorne et al. (2016) that differences will tend to be maximal in Tx, Tn, or their difference (DTR, which is not analysed further here). The Tn series are about 2°C cooler at the Stevenson Screen than the well-ventilated room with, again, the two remaining difference series coming in at about half that magnitude.

The timeseries of differences between Tn instrument pairs are less variable than for Tx, as might be expected given the lack of direct solar radiative effects on Tn (Compare lower panels of Figs 11 and 13). It also does not show a similar change in variability after 1884-85 between the well-ventilated room and the Stevenson Screen than is seen in Tx. Seasonally, the two Tn difference series with the well-

ventilated room show an apparent seasonal cycle in their differences which are smallest in Austral summer and largest in Austral winter, whereas the Thermograph – Stevenson Screen pair exhibits minimal seasonality (Fig. 14). The seasonal cycle is somewhat smaller than that for Tx. Again, for all months the differences are non-zero and thus all the differences in Tn between the three long-term series are highly significant (Table 2).

**4.2 Average temperatures**

The "average temperatures" are herein the reported averages which were preferred as the monthly diagnostic at the time, and are different from the "mean temperatures" (Tx+Tn/2), which is nowadays the standard across much, but by no means all, of the globe. Use of the average of a selection of observing hours was standard practice in very many early records (e.g. Camuffo, 2002, Bohm et al.,

2010). For the Thermograph this is stated in the metadata to be the average of the 24 hours 0LST to 23 LST. For all remaining techniques the average, when explicitly documented, is the mean of 6LST and 15LST values even though for the well-ventilated room based instrument there is evidence that observations were often taken more frequently. Early metadata until the 1890s is insufficiently detailed to determine absolutely whether the earliest averages from the well-ventilated room were 6 and 15LST

or departed from this. The Stevenson Screen only reported maximum and minimum temperatures until 1894 when average temperatures were also tabulated in the annual reports and documented as arising from 6LST and 15LST readings. For average temperatures for 1890-1900, albeit discontinuously, there are also reports from the Stevenson Screen housed Hygrometer read at the same times.

The average temperatures (Ta) series are considerably closer to one another than maximum or minimum temperatures (Fig. 15, c.f. Figs 11 and 13). Differences are generally smaller than 1°C. Hygrometer difference series are not shown given the relative brevity of that series and for presentational consistency with other similar figures. Given the similarities between the Hygrometer and Stevenson Screen series the difference series would be very small (low variance almost zero difference) for

Stevenson Screen-Hygrometer and similar to the shown differences to the Stevenson Screen for the two other set-ups. The closeness of Ta series is perhaps unsurprising given that these constituted the primary reporting metric and that, entirely reasonably, it can be implied efforts may have been made by the staff to maximise the comparability of the different set-ups deployed based upon this metric.





There is marked seasonality in the difference series for the well-ventilated room minus Thermograph in the lower panel of Fig. 15, which is readily evident in the seasonal departures shown in Fig. 16. The average temperatures from the well-ventilated room are warmer than the Thermograph in Austral winter and cooler in Austral summer. The seasonal effect also exists in the well-ventilated room – Stevenson

Screen pair although the Stevenson Screen is warmer than the well-ventilated room measurements for all except Austral mid-winter, and then not consistently so. The Thermograph – Stevenson Screen differences exhibit little seasonal structure with generally the Stevenson Screen being 0.3°C (0.1-0.5°C) warmer, although purely visually behaviour in both February and to a lesser extent October is distinct from all other months. In February the Thermograph records higher temperatures than the Stevenson

Screen, and in October the mean difference is zero. Given that solar elevation would be similar in these two months this might point to a physical effect in one or other of the instruments instead of noise. The well-ventilated room differences for each would suggest that the effect may most plausibly arise from the Stevenson Screen as the well-ventilated room-Thermograph series varies smoothly whereas the well-ventilated room-Stevenson Screen series similarly shows two peaks in November and February. In

which case vegetation shading may be the possible cause given the evidence in Fig. 4 for ample vegetation in the direction of the enclosure (Fig. 6). Later, in 1917 metadata points to RAO getting into trouble with the Sanitary Inspector for not keeping the garden and grounds in good order, in part because apparently the gardener kept falling ill. Whether these issues existed prior to this is unclear. Regardless, the scientific takeaway here is that the availability of three distinct series shows the value

that a multi-way comparison can bring over a 2-way comparison where differences can be diagnosed but never unpicked.

Formal statistical testing highlights that both the well-ventilated room-Thermograph pair and Stevenson Screen-Hygrometer pair are not statistically different under a paired t-test, whereas all other differences

are highly significant (Table 2). The Stevenson Screen and hygrometers measures are highly similar whereas the well-ventilated room-Thermograph pair non-significance relates to the cancellation of relatively large seasonal effects across the annual cycle apparent in Fig. 16.

### 4.3 Mean temperatures

Mean temperatures (Tm) here are a combination of directly calculated from Tx and Tn reports and

directly reported values to maximise the available series duration. Sometimes, for some instruments, only one or the other approach is available. With the exception of the final three years, whenever both are available, these are identical within reporting precision. In the final three years the means by which maximum and minimum temperatures were reported changed and this leads to a disconnect between the means reported in the annual report and the values inferred from the blue book maximum and minimum

temperature reports. The values based upon the blue book are used by preference in the present analysis but the effect is of the order tenths of a degree Fahrenheit so the choice should not have a substantial impact upon the present analysis. Mean temperature series are only available for the three long-standing measurement techniques and not the Hygrometer.



Mean temperature series are more similar to one another than the underlying Tx and Tn series owing to partial cancellation of marked differences between instruments for Tx and Tn (Fig. 17, c.f. Figs 11 and 13). The well-ventilated room series is consistently cooler than remaining series in Austral summer, with any differences in Austral winter being less obvious. Offsets between the different series are

somewhat larger for Tm than for Ta (lower panel of Fig. 17 c.f. Fig. 15) but are still generally within 1°C. The well-ventilated room-Stevenson screen pair carries through, with reduced magnitude, the apparent break in series behaviour in 1892 in Tx and assessed in Section 4.1.

Seasonality of the differences is marked for the two comparisons that include the ventilated room, as
was the case for Ta (Fig. 18, c.f. Fig. 16). The differences for each calendar month range over ±1°C – somewhat broader than the similar set of differences for Ta. Both comparisons involving the well-ventilated room, similarly to Ta, have a very marked seasonal cycle with differences being largest in Austral summer, consistent with the timeseries. The well-ventilated room is warmer than the Thermograph in the Austral winter, but remains cooler than the Stevenson Screen throughout the year.
The Thermograph-Stevenson Screen pair exhibits a roughly constant offset of 0.2-0.4°C across the seasonal cycle. The outliers for October and February in the Thermograph to Stevenson screen comparison in Ta no longer exist in Tm (compare lowest panels in Figs 15 and 17) pointing to a potential non-climatic effect in the 06 or 15LST instantaneous temperature in the Stevenson Screen measurements which does not impact the Tx and Tn measurements. Initial intuition might suggest
15LST as the prime candidate as radiation effects would be maximal then, but the sun would be nearly overhead making this less plausible as a transient impact than 06LST measurements which might have transient shadow effects from quite distant obstacles given the low solar elevation at that hour Without additional metadata, however, this is impossible to unpick further. All comparisons between Tm series are statistically significant (Table 2).

**4.4 The differences between mean and average temperatures for each instrument configuration**

The availability of in excess of a decade's worth of Ta and Tm measurements for the three long-standing measurement series permits an assessment of the impacts of choice of daily and monthly averaging. The norm in early records was for the monthly mean to be based upon the average of a
number of instantaneous measurements taken across the day (Bohm et al., 2010, Camuffo, 2002). Whereas many countries now calculate the monthly average as the mean of the daily maxima and minima. It is well documented that this choice can have a substantial impact upon the resulting series (Camuffo, 2002, Trewin, 2010, Bohm et al., 2010). The availability of over a decade's worth of concurrent Tm and Ta measurements for each instrumental set-up enables a quantification of the
impacts of this choice and also the sensitivity to instrumental set-up. Recall that for the Stevenson Screen and well-ventilated room Ta is the average of 06 and 15LST measures whereas for the Thermograph it is the average of 24 hourly values between 0 and 23LST.

Differences between Ta and Tm per instrument over the seasonal cycle are shown in Fig. 19. For all
three distinct measurement set-ups, Ta is consistently cooler than Tm throughout the annual cycle.



Differences always exhibit lower variance (evidenced by the dispersion of points around the median) from the Thermograph than the other two instrumental set-ups. Presumably the use of 24 hourly values in the derivation of Ta instead of 2 reduces the noise and makes the two measures more consistently equivalent to one another. The Thermograph differences are also remarkably stable across the seasonal
cycle ranging between about 0.3 to 0.5°C warmer in Tm than Ta. The well-ventilated room also tends to sit within a slightly broader range of 0.2-0.5°C warmer in Tm than Ta. There is some very slight seasonality in the well-ventilated room differences with differences being largest in Austral summer and smallest in Austral winter. The Stevenson Screen differences are much more dispersive and there is marked month-to-month variation in the differences with the median difference in November being 0°C
and October being 0.6°C. This may relate to the effects discussed in Section 4.3. There is no obvious seasonality to the pattern of differences beyond this. All three sets of differences between Tm and Ta are highly significant (Table 2).

### 4.5 The Stevenson Screen comparison

The Stevenson Screen comparison was, as noted in Section 2.2, recorded solely for 8 months during
1892. As such the substantive analyses performed for other aspects of the comparison is not appropriate and instead a tabulation of the observations is presented in Table 3. As noted in Section 2.2 the 'large' Stevenson Screen was perhaps more akin to present day Stevenson Screen sizes (although the standard screen size is not, as far as we can tell, ever explicitly documented) and the 6ft (1.8m) height screen is also closer to modern standard heights, which tend to be 1.5 or 2m rather than the 1.2m height of the
standard screen at the time. Available documentation implies that all thermometers were calibrated such that any differences will arise due to the height or housing distinctions between the three sets. Differences between the standard screen and 6ft screen are minimal until October 1892 when the 6ft screen Tx readings depart systematically to read cooler than the standard Stevenson Screen. The large Stevenson Screen Tx readings are consistently lower than the standard Stevenson Screen throughout,
whereas the Tn are consistent. Taken together, these results are suggestive of issues in the Tx with the standard Stevenson Screen (the long-standing consiguration). This may result from the relatively small screen size relative to modern day screen designs yielding estimates that are warm biased. This would certainly be supported by the large screen comparisons. There has been some limited analysis in this regard elsewhere, with Buisan et al. (2015) reporting overheating in small screens, particularly in
summer, but this is contested by Yosef et al. (2018). Regardless, the seasonal variation of the 6ft versus standard Stevenson Screen Tx behaviour would require an additional reason. Seasonality in leaf cover or vegetation shading effects is one potential explanation. The series is, however, too short to draw any firm conclusions in this regard.

## 5 Discussion

The series of measurements undertaken at the RAO are quite remarkable. Such a series of experiments, even today, is very much exceptional and undertaken only at a handful of national observatories globally. To see such a set of measurements undertaken in what was, at the time, an outpost of the then





British empire, against noted struggles of conditions, salaries, ill-health etc. shows enormous scientific fortitude. The set of measurements is very informative. The main initial scientific take-aways are as follows:

- Differences in Tx and Tn between the different instrument configurations are substantial and highly significant but are of opposite sign so tend to partially cancel for both Tm and Ta.
- There are potentially large seasonality effects which are most marked for the measurements taken in the well-ventilated room.
- Differences between Ta (average over fixed hours) and Tm (the mean of Tx and Tn) for each instrument are on the order of a few tenths of a degree and systematic with limited seasonality. Many early instrumental series used the Ta approach whereas most modern data are reported as Tm. The Tm series are systematically warmer by up to 0.5ºC.
- Because of robust and regular comparisons to a primary standard, the fact that most differences between series are highly significant points irrefutably to the non-negligible impacts of both instrumental set-ups and choice of averaging approaches.
- Assuming that the well-ventilated room and Thermograph are indicative of early measurement techniques then these recorded colder values that the Stevenson Screen measurements, at least at this location and for the particular Stevenson Screen set-up.
- However, questions around the size and elevation of the Stevenson Screen are highlighted by a short-term 3-way comparison of 3 distinct screens highlighting potential biases in Tx measurements in particular.

The differences between the well-ventilated room and the remaining instrumental set-ups, with the well-ventilated room exhibiting a muted diurnal and annual cycle, suggest that the thermal capacity of the building leads to biases in these measurements that partially cancel in the daily and annual means. Figure 4 suggests the building is at least partially stone-built, as does available blue-book metadata. The Thermograph and the well-ventilated room are both overall cold-biased relative to the Stevenson Screen. However, a short period overlap between 3 distinct Stevenson Screen configurations in the middle of the period implies that the Stevenson Screen may suffer biases, in particular in Tx arising from potentially being too small to avoid heating effects from the screen. Monthly outlier values in Stevenson Screen differences also indicate potential impacts from shading effects which impact Ta (6 and 15LST) in October and February and given the location would most logically imply issues of either early morning (perhaps more plausibly) or early afternoon shading which would tentatively be consistent with evidence of vegetation between photographic evidence in Fig. 4 and the implied positioning of the enclosure in Fig. 6.

While the comparison is highly informative around the very substantial magnitude of possible biases in early instrumental records, there exist substantial unresolved questions over how representative the findings around instrumental set-up transitions at this single locale, for this single experimental set-up, may be and hence how they may apply more broadly to the transition from early meteorological records toward Stevenson Screen-based measurements in the tropics. Firstly, it is unclear how broadly the Thermograph and / or well-ventilated room set-ups in use at the RAO were in use across former UK colonies. Based upon the Royal Society report (Royal Society, 1868) and its drive to standardisation, it



is reasonable to assume that, at least at similarly staffed facilities, it may have been encouraged and thus that the predominant transition in such cases may have been from one of a well-ventilated room or Thermograph set up eventually to a Stevenson Screen which by the 1930s was the pre-eminent measurement technique (Parker, 1994). But, equally, the RAO facility, undoubtedly a proverbial

scientific shining beacon on the hill of its time, may have been quasi-unique. Secondly, even if the techniques were broadly adopted then presumably they would have been sensitive to details such as site aspect, instrument positioning, climatological sunshine, wind, and the latitude of the site. The effect of such covariates may be large compared to any instrumental set-up effects and be unique per site. To address these issues, further efforts on metadata recovery for early tropical measurement series would

be necessary pointing to the need for renewed data rescue efforts, including efforts to rescue and manage available metadata.

The finding that the observations in the well-ventilated room, and to a lesser extent in the Thermograph screen, results in lower values than the Stevenson Screen is an interesting finding. In the seminal paper

comparing Stevenson Screen measurements to earlier observational methods, Parker (1994) finds the opposite, that earlier methods tend to record similar or warmer temperatures than Stevenson Screens. In North-West Europe these biases tend to be smaller than 0.2°C. The paper presents three tropical screen comparisons, suggesting biases can be larger in the tropics: 1) at Agra Observatory in India the mean annual temperature of a thatched shed is 0.42°C warmer than the Stevenson Screen (Field, 1920), 2) at

Colombo in Sri Lanka the mean annual temperature is 0.37°C warmer in an early felted shed than a Stevenson Screen with open bottom (Bamford, 1928), 3) at Apia, Somoa, a tropical screen (which seems to be a Stevenson Screen with a thatched roof) is 0.08°C cooler than a normal Stevenson Screen (Sapsford, 1940). Furthermore, based on a comparison of land temperatures and marine termperatures Parker estimates that tropical measurements are 0.2°C too warm for the period before the Stevenson

Screen.

Also after Parker (1994) considerable warm biases have been found in screens used before the Stevenson Screen. In two locations in Spain Brunet et al. (2011) found a warm bias in French Screens of 0.35°C compared to Stevenson Screens (only comparing screens, using modern sensors). In

Kremsmünster, Austria, Böhm et al. (2010) found a warm bias in North-wall measurements of 0.2°C compared to Stevenson Screens (again only comparing screens). Nicholls et al. (1996) found a warm bias in the Glaisher's revolving Screen in Adelaide, Australia of 0.2°C; for comparison Parker had reported on four Glaisher Screen parallel measurements in the cloudy and windy UK with on average no bias (although one of them also had a 0.2°C warm bias). A recent comparison of a Wild Screen and a

Stevenson Screen in Basel, Switzerland, found no bias in the average temperature, but a 0.2°C warm bias in the mean temperature of the Wild Screen (Auchmann and Brönnimann, 2012).

These are all early observational methods, but not well-ventilated room or Thermograph measurements. That may be what explains the difference between the current study and earlier studies on early

observations. Modelling suggests that Stevenson Screens still have a warming bias due to radiation errors (Lin et al., 2001). Thus either the well-ventilated room measurements or the Thermograph measurements may well constitute more faithful estimates of the true early temperatures in Mauritius.



Although the well-ventilated room measurements come at the expense of thermal inertia impacting their apparent ability to describe diurnal and seasonal cycles.

Significant further value would be realised if the parallel measurement data were available at daily or even sub-daily resolution alongside meteorological covariates that may allow a more physical interpretation of the causes of the differences we have found. From our inspection of the blue book series only one set of numbers is ever present for daily records and in many years it is unclear from what combination of instruments this arises. It is also possible that the thermograph readings are available for a considerable period prior to their inclusion in the blue book and annual report series as monthly averages. Unfortunately, as unfunded work, it was not possible for us to pursue these avenues of potential research further at this time, and it would require access to additional records, which it cannot be certain have been preserved. On the other hand it is very likely that blue book and annual report series entries do exist continuously. If those missing from NCEI's archives and not backfilled from the NOAA Foreign Data Library could be procured from other sources (Fig. 10) that would fill some gaps in the records recovered here.

We would caution that there is presently no direct impact of the findings herein upon the major global datasets estimating surface temperature. This is in part because data from Mauritius prior to the mid-twentieth Century have yet to be incorporated into at least some of the databases upon which these datasets are built. But more importantly, how unique the transition at the RAO is to more broadly that undertaken across the tropics is unknown. Nor is how sensitive any transition might be to site-specific covariates.

There clearly exists a very long-term and semi-continuous record between the RAO observations and observations taken at a range of earlier locations (Mahony, 2018; Section 2 and blue book entries) and there would be great value in their recovery and use; an activity ongoing under ACRE (Rob Allan, pers. comm.). Herein we have clearly only scratched the surface in uncovering the potential scientific value of this lost treasure trove of early meteorological holdings from RAO, which provide a unique window into climate in the southern Indian Ocean from 1875 to the mid twentieth Century. The subset of reports held by and imaged from NOAA NCEI contain not only meteorological observations from the island of Mauritius, but also early information on Indian Ocean tropical cyclones and many other relevant aspects of island life (Mahony, 2018; Table 1). Records were well kept and there is a rich set of contextual and societal metadata in the annual reports (less so in the meteorological reports, which mainly contain measurement system metadata). It is also possible to extract valuable additional metadata from institutional correspondence. Digitisation and exploitation of these well-managed meteorological observations and correspondence would clearly constitute a valuable addition to our knowledge of climate change in the region where exploitable directly observed climate data in the period is presently scant to non-existent. Several activities in this area are known to be ongoing (Gil Compo and Rob Allan, pers. comm.).





## 6 Conclusion

A recently rediscovered and recovered set of long-term parallel measurements undertaken over 1884 to 1903 at the Royal Alfred Observatory in Mauritius has provided valuable insights into early instrumental transitions. The principal measurements consisted of a well-ventilated room, a
Thermograph and a Stevenson Screen, supplemented by a Hygrometer and, for a short period, parallel Stevenson Screen measurements.

The instruments used were regularly calibrated against a primary standard thermometer meaning that any differences principally relate to instrumental configuration, housing and averaging effects.
Differences between instrumental configurations are large for maximum, minimum, average (average of specified hours) and mean (average of max and min) temperatures and almost all comparisons are highly statistically significantly distinct.

The findings reinforce existing literature that points to the likely presence of significant biases that may
have complex seasonal fingerprints in transitioning from early measurement techniques to modern globally standardised meteorological records. But it is unclear how representative the different configurations are of early instrumental practices nor whether site-specific effects may dominate. Thus, any broader implications for global change are not possible to be made at this time.

### Acknowledgements

The staff of NOAA NCEI who performed the indexing and imaging of the data are thanked for their efforts without which this data would remain a hidden treasure. Blair Trewin (Australian Bureau of Meteorology) and Philip Brohan (UK Met Office) are thanked for providing information on some historical instrumental practices. Rob Allan (UK Met Office) and ACRE colleagues helped source two blue books held at the NOAA Foreign Data Library to make a more complete series.

### Author contributions

S. Awe conceived of the analysis and undertook primary digitisation and initial analysis, P. Thorne undertook secondary digitisation efforts and redrafted the analysis into a paper from the MSc thesis of S. Awe, S. Noone provided Figs 1 and 3, V. Venema contributed text for the introduction around
historical analyses and parallel measurements. M. Mahony and E. Michaud provided pen sketches and history in section 2.1. T. Thorne provided Fig. 6. All authors reviewed and improved the draft text.

### Data and methods

The original imaged copies of reports are available fron NOAA NCEI and the NOAA Foreign data
library as outlined in the main text. The keyed rescued data is available at



https://doi.pangaea.de/10.1594/PANGAEA.935684 (original units) and
https://doi.pangaea.de/10.1594/PANGAEA.935683 (converted to Celsius). The IDL code (single
program) to produce all tables and plots is available at https://github.com/peterwthorne/Mauritius_code
and uses the original units version of the data files.

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



| Holdings identifier | Period of record | Temporal resolution | Observations | Additional information |
|---|---|---|---|---|
| FDL-IMG_Africa_18840101_19091231_Mauritius-AG02389-c20120625154200 | 1884-1909 annual meteorological summary reports (with some years not present) – referred to as 'blue books' | Monthly, daily and hourly (variable dependent) | Temperature, humidity, pressure, wind speed, rainfall, soil temperature | Includes tabulation of daily ozone, thunder and lightning, gales + hurricanes across the Indian Ocean from ship reports, and summaries from a number of other stations around the island. |
| FDL-IMG_Africa_18870101_19470630_Mauritius-AA00062-c20120608094100 | 1887-1974 (with some years not present) | Monthly summaries | Meteorological parameters, agricultural production and disease | Annual summary reports containing metadata and summaries of other relevant outcomes |

**Table 1. Brief summary of the two sets of image holdings archived by NOAA NCEI that pertain to the Royal Alfred Observatory. The first collection is of blue books of which two missing in the NOAA NCEI collection were sourced from the NOAA Foreign Data Library. The second is the annual report series. Neither is continuous in the NCEI holdings.**



| Diagnostic | t-test value | t-test significance |
|---|---|---|
| Tx well-ventilated room-Thermograph | -25.4113 | 0.00000 |
| Tx well-ventilated room-Stevenson Screen | -71.9583 | 0.00000 |
| Tx Thermograph-Stevenson Screen | -49.8598 | 0.00000 |
| Tn well-ventilated room-Thermograph | 32.0330 | 0.00000 |
| Tn well-ventilated room-Stevenson Screen | 62.6351 | 0.00000 |
| Tn Thermograph-Stevenson Screen | 49.0634 | 0.00000 |
| Ta well-ventilated room-Thermograph | -0.621850 | **0.535027** |
| Ta well-ventilated room-Stevenson Screen | -9.27677 | 9.66666E-16 |
| Ta well-ventilated room-Hygrometer | -7.03808 | 1.942E-10 |
| Ta Thermograph-Stevenson Screen | -12.2989 | 6.17058E-23 |
| Ta Thermograph-Hygrometer | -5.96350 | 4.20207E-08 |
| Ta Stevenson Screen-Hygrometer | -1.02132 | **0.310572** |
| Tm well-ventilated room-Thermograph | -4.28293 | 3.23407E-05 |
| Tm well-ventilated room-Stevenson Screen | -24.9421 | 0.00000 |
| Tm Thermograph-Stevenson Screen | -20.5296 | 1.4013E-45 |
| Tm-Ta well-ventilated room | -21.6161 | 0.00000 |
| Tm-Ta Thermograph | -42.1839 | 0.00000 |
| Tm-Ta Stevenson Screen | -18.9878 | 8.13292E-38 |

**Table 2. T-test results for various comparisons using a paired T-test. With the exceptions of the calculated averages for the well-ventilated room (6 and 15 LST) and Thermograph (average of 24 hours) and the Stevenson Screen and Hygrometer (both 6 and 15 LST) (both bolded) all remaining comparisons of individual temperature indicators across instruments are highly significant. Also significant are the differences between average temperatures (6 and 15LST or 24 hours) and mean temperatures (average of Tx and Tn) for the three instruments that report both.**





| Instrument / month | StS Tx | 6ft Tx | Lg Tx | StS Tn | 6ft Tn | Lg Tn | StS Tm | 6ft Tm | Lg Tm |
|---|---|---|---|---|---|---|---|---|---|
| 4/1892 | 31.2 | 31.3 | 30.7 | 21.2 | 21.1 | 21.1 | 26.1 | 26.2 | 25.9 |
| 5/1892 | 28.9 | 28.5 | 28.1 | 19.1 | 19.0 | 19.1 | 24.0 | 23.7 | 23.6 |
| 6/1892 | 26.5 | 26.1 | 25.8 | 18.1 | 17.9 | 18.0 | 22.2 | 22.0 | 21.9 |
| 7/1892 | 25.6 | 25.3 | 25.0 | 15.8 | 15.7 | 15.8 | 20.7 | 20.5 | 20.5 |
| 8/1892 | 26.3 | 26.0 | 25.5 | 16.6 | 16.4 | 16.5 | 21.4 | 21.2 | 21.0 |
| 9/1892 | 26.2 | 25.8 | 25.4 | 16.6 | 16.2 | 16.3 | 21.3 | 21.0 | 20.8 |
| 10/1892 | 28.6 | 28.0 | 27.7 | 16.9 | 17.0 | 16.9 | 22.7 | 22.5 | 22.3 |
| 11/1892 | 31.2 | 30.5 | 30.2 | 18.6 | 18.6 | 18.6 | 24.8 | 24.5 | 24.3 |
| 12/1892 | 31.1 | 30.3 | 30.1 | 20.4 | 20.5 | 20.3 | 25.7 | 25.4 | 25.2 |

**Table 3. Comparison of the three Stevenson Screen set ups undertaken in the latter part of calendar year 1892. Original values in °F have been converted here to °C. StS stands for the standard long-running Stevenson Screen measurements – taken at a height of 4ft (1.2m). 6ft is the screen mounted instead at 6ft (1.8m) closer to the typical range of screens today which tend to be at 1.5 to 2m. Lg refers to the larger screen size which is perhaps more comparable to the size of today's screens. The size of the two standard screens has not been able to be ascertained from the available metadata (see Section 2.2).**





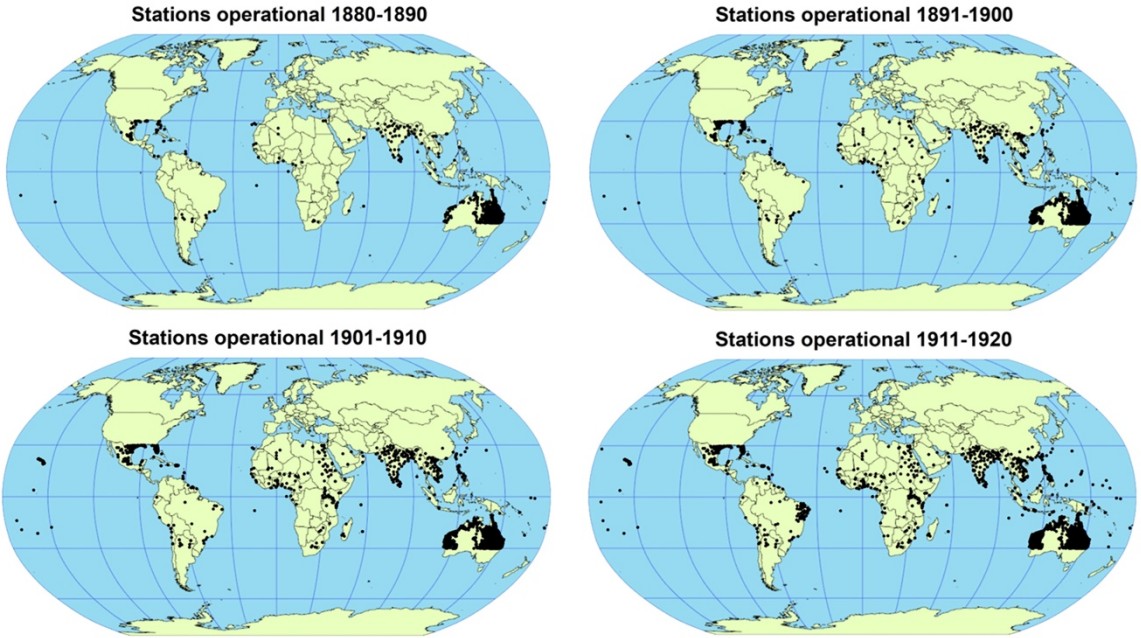

**Figure 1. Available monthly resolution temperature records between 30 degrees north and 30 degrees south over the period around the parallel measurements experiment analysed herein. Note that, to date, only non-continuous digitized data for Mauritius at Pamplemousses 1787-1974 are available in international repositories.**



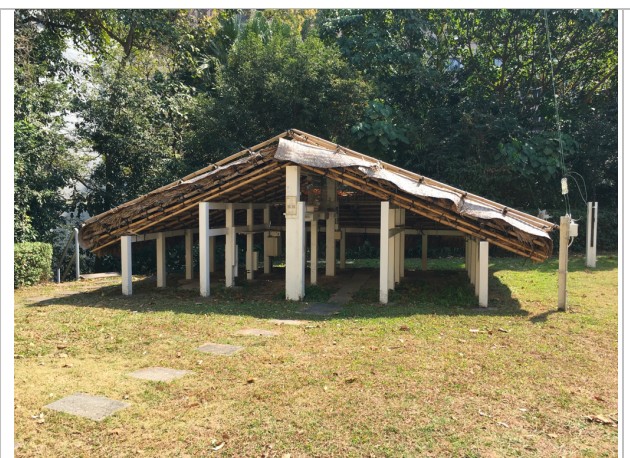

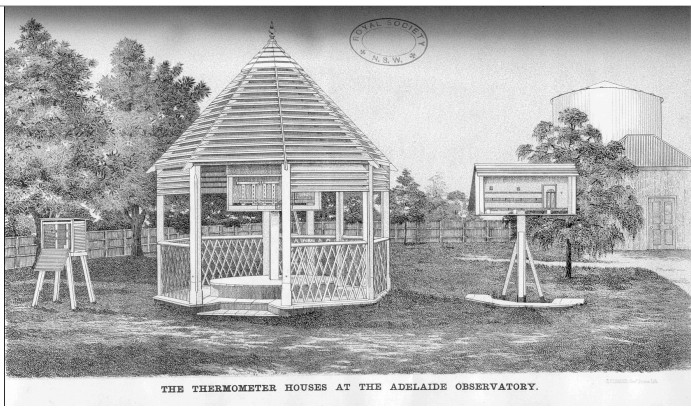

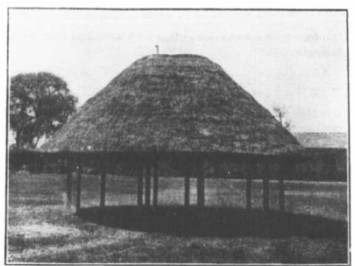

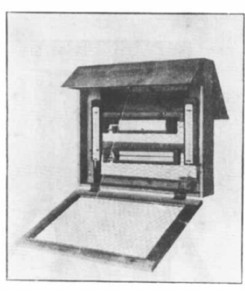

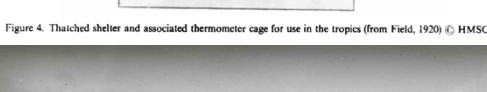

Figure 4. Thatched shelter and associated thermometer cage for use in the tropics (from Field, 1920) © HMSO

Figure 10. Tropical and Stevenson screens at Apia, Samoa (from Sapsford, 1940). © SIR Publishing, Wellington, New Zealand.

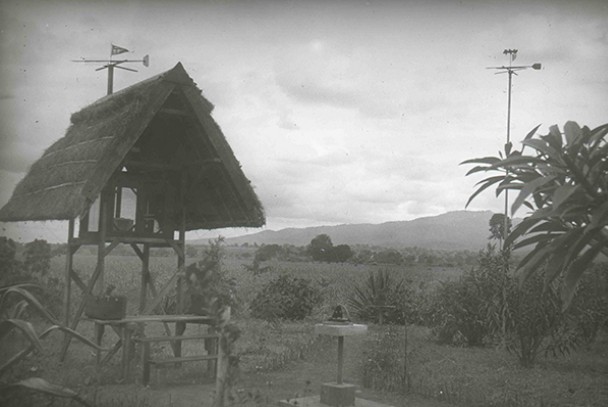





**Figure 2. Summary of a number of known early tropical / sub-tropical observational practices. Top left is a picture of the old exposure used at the Hong Kong observatory which is no longer used but still maintained (courtesy Philip Brohan). Top right shows three set-ups (a Stevenson Screen, a thermometer shed and a Glashier stand) at the Adelaide observatory which undertook 60 years of parallel measurements (courtesy Blair Trewin). The centre left (thatched shelter) and right (tropical thatched screen) are taken from Parker (1994) (Figs 4 and 10 respectively) and are the only two tropical locations with photographic evidence of instrumental set-up shown therein. The bottom left image is of the Meteorological observation station at Kizunguzi, Tanzania (Source: DWD, Archive of the Deutsche Seewarte).**





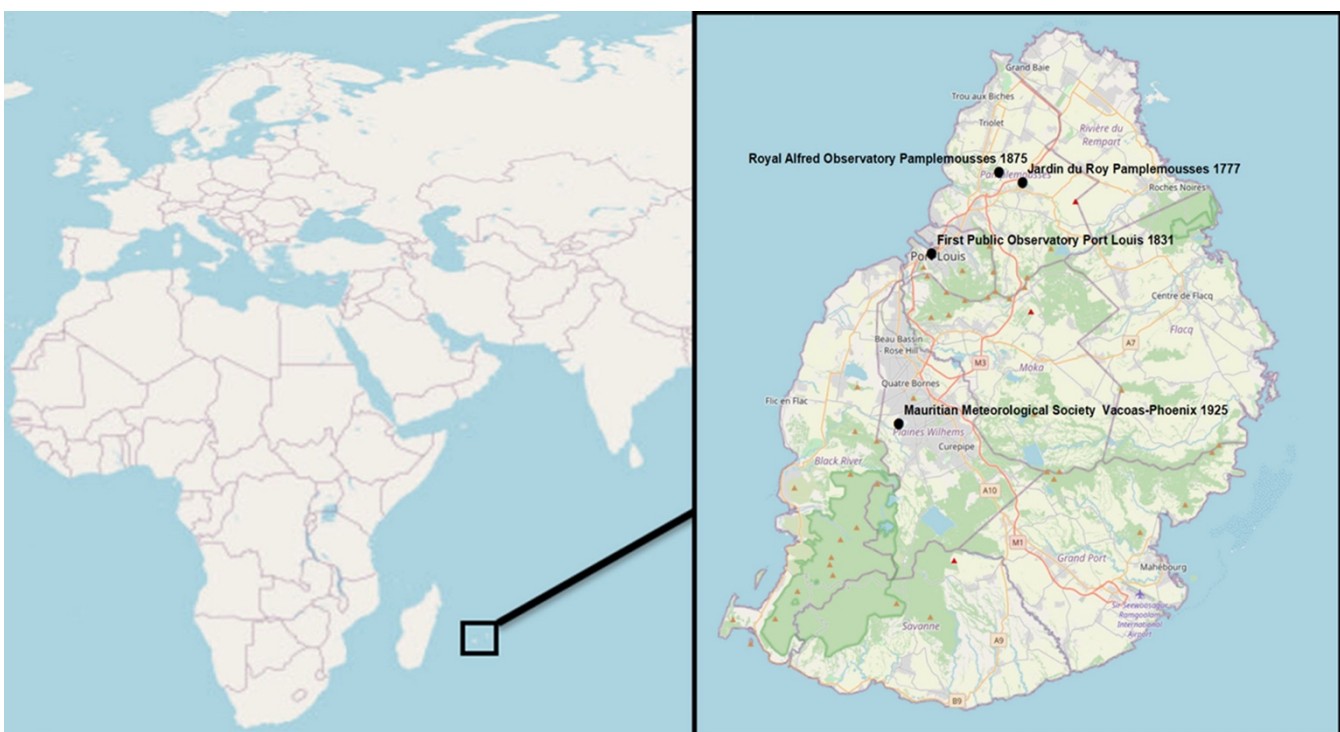

**Figure 3. Zoomed out and zoomed in imagery from Openstreetmap (© OpenStreetMap contributors 2021. Distributed under the Open Data Commons Open Database License (ODbL) v1.0) showing the position and local geography of Mauritius including superposed on the inset map the various locations referred to in the text.**






**Figure 4. Photo of the Royal Alfred Observatory building. No photos are known to exist of the actual equipment used in the parallel measurements program analysed herein. Although the metadata describes in some detail the relative positioning in later editions of the blue book series (Section 2.2, Fig. 6).**



sunshine=1) ... ... ... ... 0.022

Monthly means of Temperature in Shade, Atmospheric Pressure and Vapour-tension.

41. The following Table gives the monthly means of the temperature of the air, atmospheric pressure, and vapour-tension for the thirteen years 1875—87 and for the year 1887. The temperature and vapour-tension have been derived from observations taken at 6 and 15 hours, and the atmospheric pressure has been derived from hourly observations.

TABLE A.

| Months. | Temperature of Air. | | Atmospheric pressure at sea-level. | | Vapour-Tension. | |
|---|---|---|---|---|---|---|
| | 1875 to 1887. | 1887. | 1875 to 1887. | 1887. | 1875 to 1887. | 1887. |
| | Deg. | Deg. | Inches. | Inches. | Inches | Inches |
| January ...... | 78.6 | 78.0 | 29.948 | 29.912 | .732 | .752 |
| February ...... | 78.4 | 76.3 | 29.941 | 29.986 | .734 | .705 |
| March ......... | 77.8 | 76.6 | 29.978 | 29.958 | .728 | .747 |
| April ........... | 76.4 | 74.2 | 30.012 | 30.052 | .695 | .649 |
| May ........... | 72.8 | 71.5 | 30.088 | 30.089 | .605 | .618 |
| June ........... | 70.1 | 69.0 | 30.176 | 30.170 | .532 | .506 |
| July ........... | 68.7 | 67.6 | 30.208 | 30.216 | .508 | .502 |
| August ........ | 68.8 | 66.3 | 30.202 | 30.179 | .506 | .460 |
| September ...... | 69.9 | 68.6 | 30.198 | 30.206 | .519 | .494 |
| October ........ | 71.6 | 70.7 | 30.143 | 30.136 | .547 | .561 |
| November ...... | 74.7 | 73.6 | 30.075 | 30.098 | .603 | .578 |
| December ...... | 77.4 | 76.0 | 30.009 | 30.031 | .688 | .629 |
| Means......... | 73.8 | 72.4 | 30.081 | 30.086 | .616 | .600 |

The temperature in 1887 was $1°.4$ below the average, and below the average in every month, the greatest deviation being $2.°5$ in August. The pressure was .005 inch above the average, and the greatest deviations were .045 above the average in February and .023 below it in August. The vapour-tension was .016 inch below the average, and the greatest deviations were .059 inch below the mean in December and .020 above it in January.

**Figure 5. From the first annual summary imaged by NOAA NCEI a summary table in the front matter compares 1887 to 1875 to 1887 implying continuous measurements of air temperature over that period. Units are Fahrenheit and the image misalignment is as photographed.**




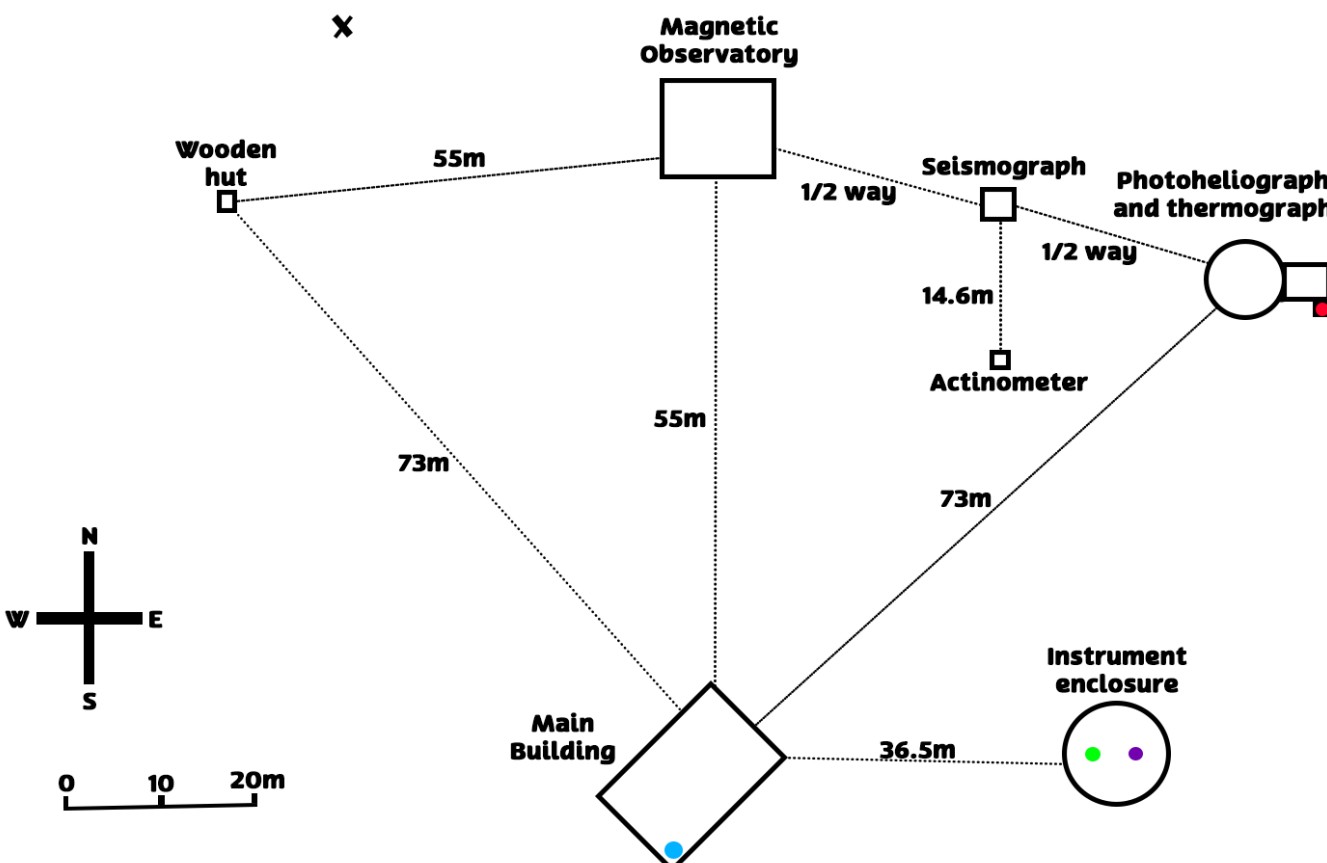

**Figure 6. Rough sketch based upon the available positional metadata in latter blue book entries of the layout of the site. Coloured dots denote the position of the 4 principal instrument configurations (light blue – well ventilated room (main building), green - Stevenson Screen (instrument enclosure), purple - Hygrometer (instrument enclosure), and red - Thermograph (Photoheliograph and thermograph building). It is probable that the wooden hut is in the right of frame in Fig. 4 placing the camera to the NNW of the main building and NE of the wooden hut (marked by X as an approximate estimate). Fig. 4 highlights considerable vegetation but the metadata on vegetation positioning and change through time does not exist.**







daily at 9ʰ.

COMPARISON OF THERMOMETERS.—All the thermometers for eye observations (except the solar radiation thermometers) are compared twice a year, near mid-winter and mid-summer, with a standard thermometer No. 701, specially constructed at the Kew Observatory. The following precautions are taken to ensure accuracy :—

The thermometers to be compared are placed in a double cistern of water on a board which is fixed at an angle of about 30° with the horizon. They are arranged symetrically on either side of the Standard with their bulbs at the same depth below the surface of the water.

The observations are commenced at sunrise, when the water is of nearly the same temperature as the air. The water is stirred before each set of comparisons, but readings are not commenced until it has become nearly still. The thermometers are read in rotation, commencing from the right and left alternately.

The observations are made with a small microscope fixed at the end of a brass tube 8 inches long, and held at right angles to the thermometer stems by a jointed arm.

The results of the comparisons in 1900 December, 1901 June, 1901 December, and 1902 July are given below, together with the correction adopted in the reductions for the year 1902. The mean of three consecutive determinations is adopted as the correction for the following six months :—

**Figure 7. Description of the biannual comparison of thermometers in the 1902 blue book imaged by NOAA NCEI, which is then accompanied by a further page of tabulated results pointing to good apparent stability of the instruments and further commentary upon assuring long-term stability of the standard via vicarious calibration against newly shipped thermometers that have been calibrated prior to shipment against the primary standard held at Kew.**



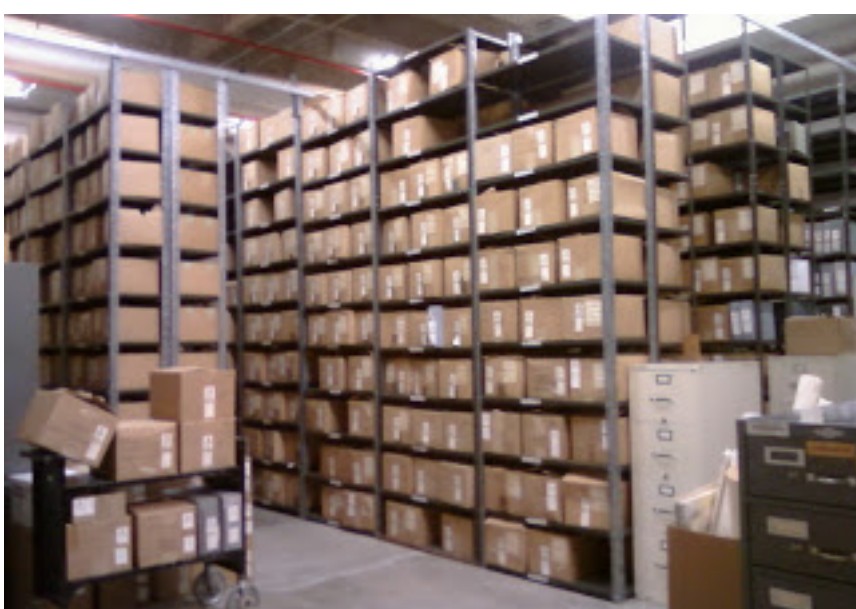

5 **Figure 8. Photo of a subset of the 6 rows of international hard copy holdings held in the basement of NCEI that were fully indexed and a small subset of which were imaged in 2012. The inventory and imaged subset can be found at ftp://ftp.ncdc.noaa.gov/pub/data/globaldatabank/daily/stage0/FDL/ and a searchable catalogue (including non-imaged holdings) is available at http://www.ncdc.noaa.gov/webartis.**



| Months. | In the Thermograph Screen. | | In a current of air between two open windows in a lofty room. | | In a Stevenson's Screen. | | a—c. | b—d. | a—e. | b—f. |
|---|---|---|---|---|---|---|---|---|---|---|
| | From 24 hourly readings. (a) | From maximum and minimum readings. (b) | From 6h. and 15h. temperature. (c) | From readings of maximum and minimum thermometers. (d) | From 6h. and 15h. temperature. (e) | From readings of maximum and minimum thermometers. (f) | | | | |
| | ° | ° | ° | ° | ° | ° | ° | ° | ° | ° |
| January ... ... | 77.9 | 78.4 | 77.3 | 77.9 | 78.6 | 78.6 | + 0.6 | + 0.5 | — 0.7 | — 0.2 |
| February ... | 77.6 | 78.2 | 77.4 | 78.0 | 78.3 | 78.4 | + 0.2 | + 0.2 | — 0.7 | — 0.2 |
| March ... ... | 76.3 | 77.0 | 76.2 | 77.1 | 77.2 | 77.5 | + 0.1 | — 0.1 | — 0.9 | — 0.5 |
| April ... ... | 76.3 | 76.9 | 76.4 | 77.1 | 77.1 | 77.4 | — 0.1 | — 0.2 | — 0.8 | — 0.5 |
| May ... ... | 71.9 | 72.1 | 72.5 | 73.0 | 72.4 | 73.1 | — 0.6 | — 0.9 | — 0.5 | — 1.0 |
| June ... ... | 68.5 | 69.1 | 69.1 | 69.8 | 68.9 | 69.7 | — 0.6 | — 0.7 | — 0.4 | — 0.6 |
| July ... ... | 66.6 | 67.4 | 67.4 | 67.9 | 67.1 | 67.9 | — 0.8 | — 0.5 | — 0.5 | — 0.5 |
| August ... ... | 66.9 | 67.6 | 67.3 | 67.9 | 67.6 | 68.1 | — 0.4 | — 0.3 | — 0.7 | — 0.5 |
| September ... | 69.7 | 70.3 | 69.7 | 70.5 | 70.2 | 70.8 | 0.0 | — 0.2 | — 0.5 | — 0.5 |
| October ... ... | 73.1 | 73.9 | 72.4 | 73.4 | 73.4 | 74.3 | + 0.7 | + 0.5 | — 0.3 | — 0.4 |
| November ... | 74.5 | 75.2 | 74.1 | 75.0 | 75.5 | 76.1 | + 0.4 | + 0.2 | — 1.0 | — 0.9 |
| December ... | 78.0 | 78.8 | 77.4 | 78.3 | 78.5 | 79.6 | + 0.6 | + 0.5 | — 0.5 | — 0.8 |
| Means ... ... | 73.1 | 73.7 | 73.1 | 73.8 | 73.7 | 74.3 | 0.0 | — 0.1 | — 0.6 | — 0.6 |

**Figure 9. Example of the annual summary sheets found in the logbooks detailing results for three independent measurement techniques aggregated to monthly averages imaged by NOAA NCEI. Temperatures were reported in Fahrenheit to a precision of 0.1 degrees. Note that each technique had averages measured from both daily maximum and minimum and native measurement resolution (hourly for thermograph screen and 0600 and 1500 local time for the others). There is no obviously available metadata pertaining as to whether maxima and minima were calculated in a consistent manner.**





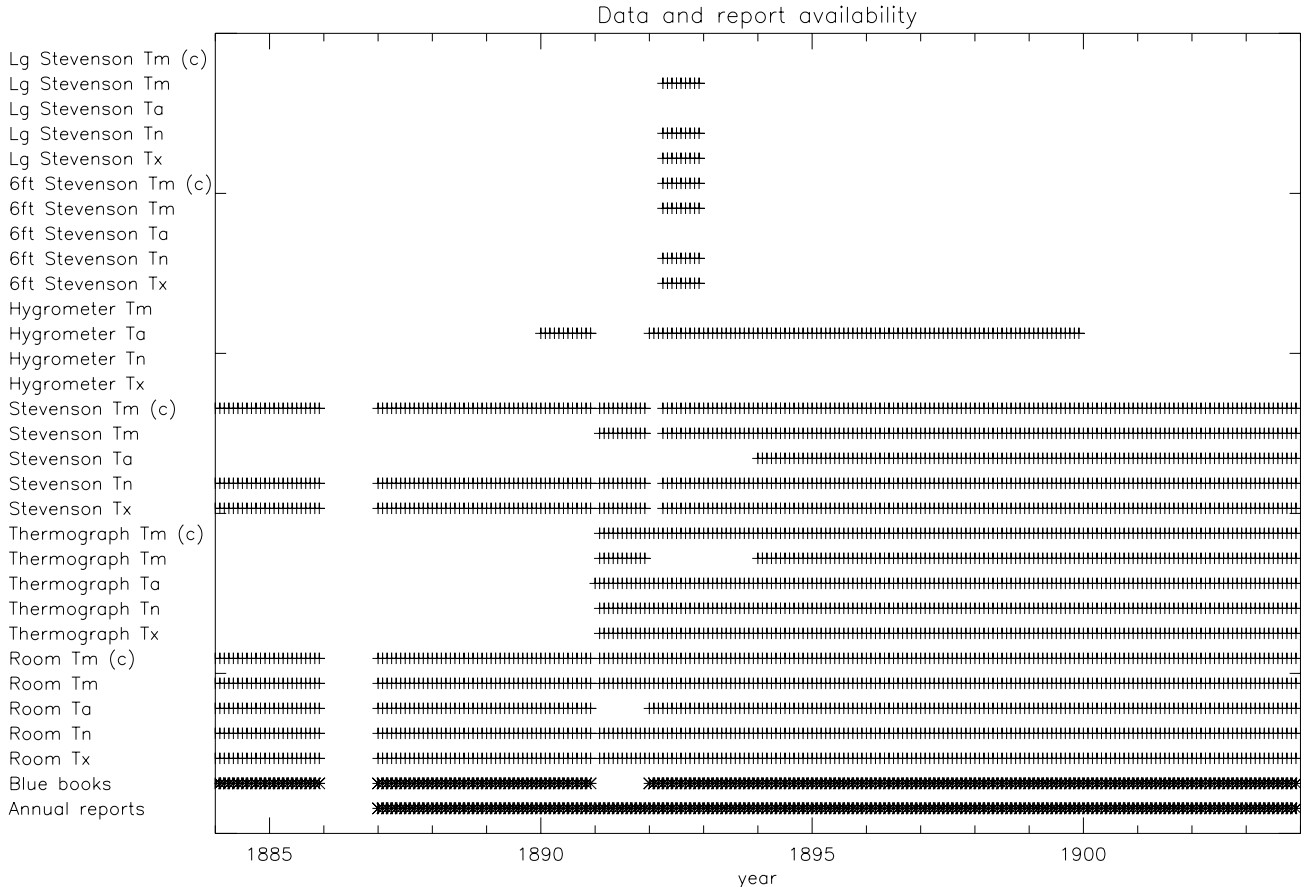

**Figure 10. Availability summary of reports (bottom two rows) in the form of Annual reports from the observatory and so-called 'Blue book' reports of meteorological and latterly magnetic observations, and then monthly reports of temperatures from the various instruments discussed in Section 2.2. Tx is maximum temperatures; Tn is minimum temperatures, Ta is the reported averages, Tm is monthly means from the average of Tx and Tn sometimes reported but also calculated for this study – denoted by (c) upon digitisation. There may also be additional records which may also be able to further fill gaps herein. Ta is calculated distinctly for the different instruments as follows: room based 6LST and 15LST; thermograph 24 hourly values; Stevenson 6LST and 15LST; and hygrometer 6LST and 15LST.**





**Figure 11. Monthly series for maximum temperatures as reported in the annual report series over the period of parallel measurements 1884-1903 from the three principal instrument series. Lower panel shows timeseries of the offsets between the measurements (in all cases first instrument minus the latter instrument).**





**Figure 12. Seasonal distribution of the differences between the different instrument pairs to the extent each series is available (Figs 9 and 10). Each individual value is specified by a cross and the median of all values is denoted by the solid horizontal line for each calendar month. Vertical axes are kept identical between panels for comparison purposes.**





**Figure 13. As Fig. 11, but for minimum temperatures.**





**Figure 14. As Fig. 12 but for Tn. Axes ranges are identical between panels but differ from those in Fig. 12.**





Figure 15. As Fig. 11, but for monthly average temperatures (Ta) that extend for longer and are more contiguously available from all four techniques (including the hygrometer). For all techniques except the thermograph Ta is calculated from the mean of observations at 6 and 15LST. For the thermograph it is the mean of the 24 hourly observations. The hygrometer series is relatively short and it is shown only in the top panel. Significance of differences are assessed in Table 2 for this instrument.

## Average temperature differences

ventilated room − thermograph

ventilated room − stevenson screen

thermograph − stevenson screen

**Figure 16. As Fig. 12 but for Ta. Axes labels are identical across the three panels but differ from those in Figs 12 and 14.**







**Figure 17. As Fig. 11, but for mean temperature (the average of Tx and Tn). To make the record as complete as possible a combination of directly reported and self-calculated estimates has been used. Where both numbers exist they match entirely except for the period 1901-1903 when the method of tabulation of averages changes introducing a disconnect of the order tenths of a degree Fahrenheit between the series in the blue book and the annual report prior to their conversion here to °C.**



## Mean temperature differences



**Figure 18. As Fig. 12 but for Tm. All axes are the same range but differ from those given in Figs 12, 14, and 16. As in Fig. 17 the series utilises a combination of self-calculated and reported values for Tm.**



Figure 19. As Fig. 12 but for the differences between reported average and mean temperatures for each individual instrument configuration. All axes ranges are identical to aid comparability.