# Peer review of "Insights from 20 years of temperature parallel measurements in Mauritius around the turn of the $20^{th}$ Century"

_Climate of the Past, 2021_

## Author Comment (AC1)

**Response to review by Dr. Stephen Burt**

We thank Dr. Burt for his careful evaluation of the manuscript. We include below on a point-by-point basis responses in red to the points made in review.
* * *
**Overall impression**

This is a well-researched and useful piece of work. As the paper describes, multi-instrument comparisons are few and far between, particularly for the tropics in the 19thC, and placing the insights resulting from this analysis on record will be helpful to others attempting to unravel any biases in other similar record types. I have no hesitation in recommending acceptance subject to minor revisions. Some suggestions are given below.

Thanks for your positive assessment of the paper

**Minor comments and suggestions**

Fig. 2 caption: the thatched screen at Hong Kong is, contrary to the caption, still very much in use today! - certainly it was on my most recent visit in October 2018.

Thanks – you are correct. What we meant to imply is that although maintained it no longer constitutes the operational measurement series. We have changed used to operational in the caption to avoid this source of evident confusion.

Fig. 3: a scale on the island map would be helpful.

A scale has been added as suggested.

Fig. 4: A date for the photograph, even approximate, would be helpful. It appears to have been taken from a book - perhaps include the source as a reference, or at least the date of publication of the book?

That image is actually a postcard, making it difficult to date without a hard copy, but our best guess is that it's from around 1909. Magasins Réunis was a chain of department stores, so they would count as the publisher here. A very similar photograph (we think identical but with a different crop) was published in Walter's 1910 *Sugar Industry of Mauritius*, so it's certainly no later than that. Unfortunately we don't seem to have a good quality scan of Walter's version though. We have amended the figure caption to give some of this detail.

Figs 11 and 15: spelling error in legend, should read 'ventilated'

Thanks for spotting this. Corrected in all relevant figures

page 20 line 21, Samoa not Somoa

Thanks. Corrected

Page 9 ff: the term 'thermograph' in this context clearly differs from the modern usage, viz. a small portable instrument housed in a Stevenson screen, with a bimetallic coil as the sensor, recording on daily or weekly paper charts wound around a clock drum. Perhaps it may be advisable to make this distinction clear at first usage. I would also suggest 'thermograph' (lower case t) in place of Thermograph.

We have clarified the distinction when the Thermograph metadata is introduced as that makes it easiest for a reader to draw the distinction in methods. We prefer to retain methods of observation as capitalised proper nouns for ease of readability.

A minor point, but the thermograph is described as being by Adie on p8 line 36, but by Hicks on p10 line 1. Perhaps one was a subsequent replacement?

Thanks for catching this. Indeed, this mismatch is intriguing. Once a Thermograph measurement series is available the metadata consistently refers to as Hicks and not Adie and yet the early metadata about shipment of a Thermograph consistently refers to an Adie model. We have clarified this by extension of the paragraph that was on p.8 in the discussion version of the paper.

There are various references in the text to differing sizes of Stevenson screens. It should perhaps be borne in mind that the modern 'large' or double-width Stevenson screen did not appear until well after the period described in this paper - they were introduced in World War I to allow autographic instruments to be sited alongside conventional thermometry - and that screen size differences referred to may have been less than those between modern 'standard' and 'large' thermometer screens. Two contemporary papers (Mawley 1884, Council RMetS 1884) provide details and dimensions of the (slightly smaller) original pattern of Stevenson screen, and the larger model approved following trials by the RMetS Thermometer Screen Committee in 1883-84, where the dimensions differ by only 5 cm or so. It could be that these are what is meant by the 'small' and 'large' screens. From Mawley 1884: 'old' screen W 16 in, D 9 in, H 16.5 in; 'new' screen W 18 in, D 11 in, H 16.5 in.

Thanks for this detailed information, some of which we have directly incorporated into the paper. The metadata in the report series suggests that this is distinct from the Mawley documented analysis in that there is a far more considerable distinction between the screen sizes – perhaps more akin to the difference between what may be considered a modern screen and the small screen as discussed in Mawley. We have inserted comment to this end in an appropriate point of the revised manuscript.

The larger 'thermograph' screen is referred to as a Kew pattern (although it is likely that it was first developed at Oxford's Radcliffe Observatory about 1849). At Kew it became known as the North Wall Screen, with the bulbs of the thermometers sitting outside the building wall within a large louvred screen, the stems recording via photographic paper on a drum mounted inside the building. (This remained in use until the Observatory was closed in 1980.) The Mauritius setup sounds very similar; there is a contemporary plate in Anon 1892 and photographs of the Kew screen in Drummond (1947, Plate II) and in Galvin (2003, Fig 2). Inclusion of one of these images may be worth considering. There is no doubt, however, that the size of the thermometer bulbs and stem required would have increased the response time considerably, and this factor when combined with thermal inertia of the building would have resulted in lower maxima and higher minima, and thus reduced DTR, when compared to a Stevenson screen record in the open air.

Thanks for this additional information. We have added much of it to the revised manuscript. Given the length of the manuscript we feel we cannot add a figure as suggested but do refer the reader to the suggested materials given which we agree greatly aid reader interpretation.

The radiation errors of large Stevenson screens in a subtropical desert climate were found to be less than other types of screen in a WMO trial in Algeria (Lacombe, 2011). In mid-latitudes, it is well-known that strong solar radiation can result in considerable heating of the louvred sides of the screen and result in screen temperatures warming over 'true' air temperature (as measured by, for instance, an aspirated sensor); but in tropical latitudes, with higher or overhead solar angle, the radiation errors did not appear to be as great as might have been expected, probably owing to shadowing of the louvres from the screen roof. Midlatitude screen comparisons are unlikely to be representative of tropical sites for this reason.

Thanks. We use this information directly in the revisions to provide a valuable caveat.

Throughout: metadata are plural not singular. Suggest omission of all imperial dimensions, leaving only metric units, unless there is a clear case for retention.

We have tried to be more consistent in using metadata as plural. We prefer to retain original units where relevant while also providing metric equivalents. That way the trace is more direct and also if we have introduced a conversion error the reader can identify and correct for this.

References

Anon, 1892: Weather watchers and their work. London: The Strand Magazine, 3, 182-189.

Council of the Royal Meteorological Society, 1884: Report of the council for the year 1883: Appendix 1, Report of the thermometer screen committee. Quart. J. Royal Meteorol. Soc., 10, 92-94.

Drummond, A. J., 1947: Kew Observatory. Weather, 2, 69-76.

Galvin, J. F. P., 2003: Kew Observatory. Weather, 58, 478-484.

Lacombe, M., D. Bousri, M. Leroy et al, 2011: Instruments and observing methods report no. 106: WMO field intercomparison of thermometer screens/shields and humidity measuring instruments, Ghardaïa, Algeria, November 2008 – October 2009. Instruments and Observing Methods Report No. 106, WMO/TD-No. 1579, Geneva, Switzerland.

Mawley, E., 1884: Report on temperatures in two different patterns of Stevenson screens. Quart. J. Royal Meteorol. Soc., 10, 1-7.

All suggested references have been added

Stephen Burt

Department of Meteorology, University of Reading

30 November 2021

---

## Author Comment (AC2)

**Response to review by Dr. Linden Ashcroft**

We thank Dr. Ashcroft for her careful evaluation of the manuscript. We include below on a point-by-point basis responses in red to the points made in review.

_________________________________________________-

This paper describes and explores an incredible parallel dataset for the Indian Ocean nation of Mauritius. The dataset presented in this paper is rare in its age and location, and the authors have done a service to the community by bringing it to light. While the analysis of the data is simple, it provides a useful new study of parallel observations in a region severely lacking in any historical weather data, let alone multiple simultaneous records.

Thank you for this nice synopsis of the work and its importance.

The paper would be appropriate in a data journal such as Earth System Science Data or Geoscience Data Journal, but I believe the paper is suitable for Climate of the Past given the rarity of the record. I therefore recommend it be published once the authors have considered the suggestions below. Most of my suggestions are about improving the readability of the article so as many people can learn about this dataset as possible.

Thanks. We did consider the alternative journals you suggested above when we were preparing the manuscript but in the end decided that given the relative antiquity of the recovered records and the amount of metadata (both about the scientists and the measurement series) associated that it was most germane to a Climate of the Past audience.

1. My main comment on this manuscript is that is it very long. The sections on the key personnel of the RAO (in section 2.1) and the detailed metadata from the 1899 blue book (in section 2.2) could be placed in a supplementary section, with a one paragraph summary given in the main text instead. This would allow readers to focus on the key findings of the data analysis and explore the history of various elements if they are really interested.

Thanks. We have reflected upon this but really feel that we do need to do justice to the context of the measurement series and fear that placing this material in an appendix reduces its importance. We wanted the piece to reflect both the series but also the remarkable context in which these early scientific pioneering efforts were undertaken by a truly remarkable group of individuals. This in part was why we chose Climate of the Past where we felt we could provide that balance between the context and the results which we could not do in ESSD or GDJ. We therefore would

prefer to retain this in the main text so that we can fully appropriately recognise these forgotten scientific heroes and the experimental details held in these records and we feel this is broadly in keeping with the journal remit to do so.

2. Similarly, the introduction could be tightened to tell the story of these valuable observations more directly. The current narrative of Paris agreement > early instrumental biases > homogenisation techniques > tropical data scarcity > necessity of parallel observations could be shortened to 'we need more data > the best kind of data are data we can trust > parallel obs allow us to have more confidence in early data but they are rare > we found some'! These ingredients are in the paper now (mainly section 1 and 3), but they could be rearranged.

Thanks. We have redrafted the introduction to improve this narrative aspect without losing aspects we felt were still important to justify why this analysis was important from a policy perspective.

3. Finally, I don't mean to be that person who mentions their own work in a review, but given the Southern Hemisphere location, work we've done in Australia looking at simultaneous changes in observation practises might be relevant (Ashcroft et al., 2012) if you want to keep some of that discussion in your introduction. Our recent publication on the parallel observations in Adelaide will also be useful (Ashcroft et al., 2021). In particular, the references to Blair Trewin personal comms could be replaced with this latter citation, as that paper contains all the details. It's encouraging your analysis in the tropics is similar to our assessment of the mid-latitude Adelaide dataset, particularly in terms of greater seasonality in the Tx differences compared to Tn.

Thanks. We only learnt via Blair Trewin that Ashcroft et al had been published a week after the discussion paper was published and hoped that a reviewer would note this so that we could include it. We now include the papers in the revision.

**Minor/typographical**

Throughout: Extra commas throughout the manuscript would improve the readability of this paper, as I found I tripped up on the start of many sentence. E.g. Page 3, line 16 "when such biases change, they...', page 3, line 19 "prior to use in climate applications, these records...", page 13, line 17 "As the work progressed, the...", page 18, line 15 "As such, the substantive..."

Reread and attempts made to improve grammar throughout

Throughout: 'data' and 'metadata' are plural

Checked and adjusted as necessary

Throughout: austral is lower case

Corrected throughout

Abstract line 12: within roughly 80 metre radius of each other?

Useful clarification. Implemented.

Page 3, line 26. The new sentence on this line could be the start of a new paragraph

Done

Page 4 line 28: Shamefully, I've never heard the Stevenson Screen being called a Cotton Region Shelter before! Is that a US name? A reference would be good here.

Yes, it's the US name. We have added Quayle et al here where it is discussed in the context of the transition of the COOP network from these to MMTS sensors and associated impacts. This may help others who have similar concerns.

Page 4, line 35–38: this long sentence could be broken into two.

Accepted. Split into two

Page 6, line 9–11: you could remove this sentence as its message is clear from the previous one.

Accepted. Removed.

Page 6, line 18: you mention Mr Cere's many vocations, but then call him a scientist for the first time when he began publishing. That confused me a bit – did he train as a scientist and then start publishing, or call himself a scientist in his publications?

There is some confusion here potentially. The many vocations are associated with Lislet Geoffroy and not Mr. Ceres. In that context it is unclear whether changes are required and so we have left this passage as is.

Page 11, first paragraph: Figure 10 should be mentioned here, as this timeline is hard to follow just by reading it.

Agreed. Figure 10 reference added.

Page 13, final sentence: I think you could make more of the fact that the data are available. It feels buried in this section and would be better in the conclusions. In revising the introduction you might find that Section 3 is not needed.

Data availability is given also after the conclusions per journal guidelines. We prefer to retain section 3 as a distinct section.

Page 14, line 4: Is there any reason why you have used 'were' instead of 'if' at the start of this sentence? Using 'were' makes that sentence harder to follow in my opinion.

We prefer to retain were here but have added 'to be' later in the sentence in the hope that this clarifies the intent of this.

Page 14, line 16: what do you mean by change in seasonality? An extra bit of info here saying that the differences are reduced during the cool part of the year compared to later observations would be helpful.

We have expanded this text accordingly to try to provide better quantification.

Page 14, line 19–20: I found this sentence hard to follow. "For the shift around 1892, there is for most of 1891 and thereafter also the Thermograph available permitting a 3-way comparison." Can you rephrase it?

We have attempted to clarify

Page 14, line 23: you talk about a major cyclone in April 1892 like we know all about it, but I didn't see earlier reference to it in the article. Perhaps you could say 'a major cyclone in April 1892' and/or provide a reference?

Added 'documented in the reports' as this insight arose from the reports from which these data arose

Page 15, line 18: Can you provide a reference for the statement about mean temperatures? There is probably guidance around this in (World Meteorological Organization, 2011)

A simple reference is unfortunately frustratingly hard to come by. The WMO 2011 document referenced in the review is unfindable on the new WMO website (at least when we looked over week of 24$^{th}$ Jan 2022). We know that WMO does not mandate the use of Tx+Tn/2 but most stations (if not countries) do follow this approach. This is one of these things that is 'known' in the community but not adequately documented. The closest we can get is the WMO report on calculation of climate normals () but this solely supports the part of the sentence around heterogeneity in approach and does not adequately support the main contention around Tx+Tn/2 being the predominant approach. There are copious web-based statements to support the latter point and e.g. very murky grey literature but nothing that we feel can be cited. We include WMO (2017) as above to support what we feel it can but note that it only supports adequately parts of the sentence.

Page 16, line 33: do you know why the reporting changed in those final three years?

No, no rationale or justification is given. This is now noted in parentheses as others may have similar questions. Thanks for raising this.

Page 19, line 3: you don't need to say 'as follows'

As follows has been removed.

Page 19, line 36–40: this sentence would be more powerful broken into two.

We have broken the sentence into two.

Page 20, lines 4–5: wouldn't there be more written about the RAO if it was a shining scientific beacon of its time? A reference to Mahony 2018 here might remind the readers that there are more explorations of the output from this remarkable observatory.

Agreed, added

Page 20, final sentence: I'm a bit surprised by this statement, given that the Stevenson Screen is the world standard for exposure and has been for a long time. It's not perfect, but it's generally considered the best we can do. It would be worth clarifying which temperatures you think the room and Thermograph observations are more accurate for. In the next sentence you state that these observations sacrifice accuracy of daily and seasonal cycles, so are you saying they are most accurate for annual observations, rather than daily on monthly extremes?

As noted by Dr. Burt in his review these Stevenson screens were smaller than modern exposures. We amend text in various places including here accordingly.

Page 21, line 24: I'd remove the word very here, the sentence says the same thing without it.

Removed 'very'

Table 1: Can you list the years that are not present in each holding in this table? That will make it more useful to future researchers exploring these data

The information as it pertains to the present study is given in Figure 10. We now include in the table caption a forward reference to that figure. We did not perform an inventory outside these periods but did note the overall span of the records.

Table 2: Thick lines between each variable (Tx, Tn etc) would improve the readability of this table. I'm also not sure you need so many significant figures in your t-test results.

We added different shadings to differentiate the different elements. The table entries have been reduced to 2d.p. in all cases to avoid undue precision.

Table 3: Similarly, thick vertical lined between each comparison would help this table

Similar to Table 2 we suggest using different shading to denote this here.

Figure 2: The Adelaide observatory image comes from the Royal Society of New South Wales.

Attribution changed

Figure 4: A comma needs to replace the full stop before 'Although'

Changed accordingly

Figure 6: How hard would it be to change the coloured dots to each have a different shape, for colourblind readers (or those who printed the paper out in black and white)?

The software that was used to generate this figure has been deprecated so we prefer to maintain as is.

Figure 7: I'm not sure you need this figure – it could go as an appendix

We prefer to retain this figure for provenance of the discussion in the main text.

Figure 8: Again, not sure this figure is needed, particularly as it is low resolution

We think there is value in retaining this figure to illustrate the scale of the original hard copy records at NOAA NCEI to raise general awareness of the potential value in further investigation. We feel this is of likely interest to a CP journal audience.

Figure 11, 13, 15: Ventilated is spelt incorrectly in the legend. It would also be better to use 'differences' rather than 'offsets' in the caption, to be consistent with the axis label. Finally, is there a way you could differentiate the lines on the chart for colourblind readers? Different thicknesses or line types perhaps?

We have corrected the spelling. We played around with various linestyle and colour options in producing the original draft because of concerns we had. over accessibility. We were unable to find a more satisfactory option than present. The use of e.g. dashed lines causes issues of interpretation and we could not find a set of colours that may work better than those given. We tried to map these as closely as we could to various colour-blind friendly solutions.

Figure 12, 14 and 16: This is monthly distribution rather than seasonal

Corrected

Figure 15: There looks to be an outlier in the Hygrometer data in 1894 - can you comment on this?

Thanks. We have checked this and found a transcription error that has now been corrected and the data holdings and figure reissued.

Ashcroft L, Karoly D, Gergis J. 2012. Temperature variations of southeastern Australia, 1860–2011. *Australian Meteorological and Oceanographic Journal*, **62**(1): 227–245.

Ashcroft L, Trewin B, Benoy M, Ray D, Courtney C. 2021. The world's longest known parallel temperature dataset: a comparison between daily Glaisher and Stevenson Screen temperature data at Adelaide, Australia, 1887–1947. *International Journal of Climatology*, (December 2020): 1–18. https://doi.org/10.1002/joc.7385.

World Meteorological Organization. 2011. *Guide to Climatological Practices*. Geneva. Available at https://library.wmo.int/pmb_ged/wmo_100_en.pdf.

---

## Author Response (AR2)

Dear authors,

I thank you very much for the revised version of your paper. You have either fixed or explained why you did not change the issues mentioned by the referees. The paper is therefore accepted for publication with the request to check the capitalization of some expressions. Of course, these are only smal details, which is why this can be implemented with "technical corrections".

Once again I sincererly thank both referees.

Kind regards,

Chantal Camenisch

Non-public comments to the Author:
Dear authors,

I think "century" should be lower case in p. 6, line 22. "Blue book" appears in the paper in both upper and lower case. I suppose it should be capitalized in most cases. Possibly, on the other hand, "thermograph" should perhaps be lowercase on p. 10, line 13 (it makes of course sense to me that you capitalize the term in most cases). On the other hand, I suspect that "screen" should be lowercase on p. 11, line 32 and 35.

yours sincerely,

Chantal

Thankyou for your careful proofing. We have made suggested changes in the revision as well as addressing the request not to use colour cells in the tables 2 and 3 from the copyeditors (although we would note that this does somewhat reduce readability).